# The conserved AAA ATPase PCH-2 distributes its regulation of meiotic prophase events through multiple meiotic HORMADs in *C. elegans*

**Anna E. Russo[1], Stefani Giacopazzi[1], Alison Deshong[1], Malaika Menon[1], Valery Ortiz[1], Kaori M. Ego[2], Kevin D. Corbett[2], Needhi Bhalla[1]***

1 Department of Molecular, Cell and Developmental Biology, University of California, Santa Cruz, California, United States of America, 2 Department of Cellular and Molecular Medicine, University of California, San Diego, California, United States of America

* nbhalla@ucsc.edu

**Data Availability Statement:** The data for all figures is provided as an attached excel file labeled Supplementary Data File.

## Abstract

During meiotic prophase, the essential events of homolog pairing, synapsis, and recombination are coordinated with meiotic progression to promote fidelity and prevent aneuploidy. The conserved AAA+ ATPase PCH-2 coordinates these events to guarantee crossover assurance and accurate chromosome segregation. How PCH-2 accomplishes this coordination is poorly understood. Here, we provide evidence that PCH-2 decelerates pairing, synapsis and recombination in *C. elegans* by remodeling meiotic HORMADs. We propose that PCH-2 converts the closed versions of these proteins, which drive these meiotic prophase events, to unbuckled conformations, destabilizing interhomolog interactions and delaying meiotic progression. Further, we find that PCH-2 distributes this regulation among three essential meiotic HORMADs in *C. elegans*: PCH-2 acts through HTP-3 to regulate pairing and synapsis, HIM-3 to promote crossover assurance, and HTP-1 to control meiotic progression. In addition to identifying a molecular mechanism for how PCH-2 regulates interhomolog interactions, our results provide a possible explanation for the expansion of the meiotic HORMAD family as a conserved evolutionary feature of meiosis. Taken together, our work demonstrates that PCH-2's remodeling of meiotic HORMADs has functional consequences for the rate and fidelity of homolog pairing, synapsis, recombination and meiotic progression, ensuring accurate meiotic chromosome segregation.

## Author summary

Sexual reproduction relies on production of gametes, such as eggs and sperm, which are produced during meiosis. During this specialized cell division, chromosomes replicate, pair with their homologs, undergo synapsis and finally undergo recombination, all of which are required for correct meiotic chromosome segregation. How these events are coordinated with each other, and the cell cycle, to ensure that they occur with fidelity is

**Funding:** This work was supported by the NIH (grant number T32GM008646 [A.E.R], R35GM141835 [N.B.] and R35GM144121 [K.D.C.]). Some strains were provided by the CGC, which is funded by NIH Office of Research Infrastructure Programs (P40 OD010440). The Advanced Light Source is a Department of Energy Office of Science User Facility under Contract No. DE-AC02-05CH11231. The ALS-ENABLE beamlines are supported in part by the National Institutes of Health, National Institute of General Medical Sciences, grant P30 GM124169. The funders had no role in study design, data collection and analysis, decision to publish, or preparation of the manuscript.

**Competing interests:** The authors have declared that no competing interests exist.

not well understood. In previous work, we showed that the conserved enzyme PCH-2 coordinates homolog pairing, synapsis and recombination in the nematode *C. elegans*. Here, we show that PCH-2 accomplishes this coordination by distributing its regulation of these events through different members of the meiotic HORMAD protein family: PCH-2 controls pairing and synapsis through HTP-3, recombination through HIM-3 and meiotic progression through HTP-1. Moreover, this distribution may explain why there are so many meiotic HORMADs in *C. elegans*. Finally, we provide a model for how PCH-2 may be modifying meiotic HORMADs through its enzymatic function, molecularly explaining its role in pairing, synapsis, recombination and meiotic progression.

## Introduction

All sexually reproducing organisms depend on the specialized cell division called meiosis, which produces haploid gametes such as sperm, eggs, and pollen. During meiosis, cells combine one round of DNA replication with two rounds of chromosome segregation, all of which are tightly regulated to ensure that gametes inherit the correct number of chromosomes. Errors in meiosis can lead to aneuploid gametes that cause infertility, birth defects and miscarriages [1]. Therefore, understanding how meiotic events are coordinated has important implications for reproductive health.

During prophase I of meiosis, chromosome pairs or homologs undergo a series of events known as pairing, synapsis, and crossover recombination. First, homologs find each other and pair within the nucleus. Next, a molecular structure called the synaptonemal complex (SC) assembles between homologs and stabilizes homolog pairing in a process called synapsis. Finally, homologs undergo crossover recombination, which serves two important purposes. First, it shuffles alleles to create new genetic combinations that can be acted upon by natural selection. Second, it physically links homolog pairs, which is required for homologs to properly remodel, orient on the spindle, and segregate correctly during the first meiotic division.

Despite being described as a linear pathway, the processes of pairing, synapsis and recombination between homologous chromosomes are highly dynamic and can occur simultaneously, requiring active coordination. Further, these events can also initiate inappropriately between non-homologous chromosomes and therefore need to be corrected. We have shown that the highly conserved AAA+ ATPase, Pachytene CHeckpoint 2 (PCH-2), contributes to both the coordination and correction of meiotic events [2,3]. However, how it carries out these roles is currently unclear.

PCH-2, and its mammalian ortholog TRIP13, have been shown to act through a family of proteins that contain a HORMA domain (HORMAD proteins) to mediate a variety of chromosome behaviors, including meiotic interhomolog interactions. HORMADs were named after the first proteins in this family to be discovered in budding yeast: Hop1, Rev7, and Mad2 [4] and meiotic HORMADs are essential to organize meiotic chromosomes for the meiosis-specific events of homolog pairing, synapsis, and recombination [5]. Despite having essential roles in meiotic prophase, meiotic HORMADs differ greatly across eukaryotes in the number of family members they contain. For example, *S. cerevisiae* and *S. pombe* each have one meiotic HORMAD [6,7], mice and *Arabidopsis* have two [8–10] and *C. elegans* has four [11–14]. In both mice and plants, one meiotic HORMAD plays a more integral role in the events of pairing, synapsis, and recombination than the other, indicating some specialization of function [9,10,15–18]. This specialization is also observed in *C. elegans* among the three essential HORMADs [19]. HTP-3 is responsible for DNA double-strand break formation [12] and REC-

8-mediated sister chromatid cohesion [20]. HIM-3 promotes synapsis [19] and homolog access to ensure repair off the homolog versus the sister chromatid during crossover recombination [21]. HTP-1, in addition to contributing to homolog access along with HIM-3, also controls the timely initiation of SC assembly once homologs are paired properly [11,13] and the protection of sister chromatid cohesion during the first meiotic division [22]. The fourth meiotic HORMAD, HTP-2, is not essential and is largely redundant with HTP-1 [13,22].

Meiotic HORMAD proteins have a characteristic N-terminal HORMA domain that allows HORMADs to adopt two different conformations: a closed conformation and an open/unlocked conformation (reviewed in [23]). In the closed conformation, a peptide on a binding partner called the "closure motif" binds the core of the HORMAD's HORMA domain. This interaction is stabilized by a C-terminal region of the HORMA domain called the safety belt, which wraps around the HORMAD to lock the binding partner's closure motif against the HORMA domain core. Conversion of the HORMAD from the closed to open conformation occurs when the PCH-2/TRIP13 binds and partially unfolds the extreme N-terminus of the HORMA domain, resulting in a structural rearrangement that releases the bound closure motif and causes the safety belt to occupy the closure motif interaction site. Biochemical and structural studies show that PCH-2/TRIP13 is directly responsible for the conversion of closed conformers to open conformers for the mitotic HORMADs Mad2 [24] and Rev7 [25], where this remodeling relies on a direct interaction between PCH-2/TRIP13 and the N-terminus of the HORMAD [26,27].

Conformational switching has also been shown to be essential for meiotic HORMAD function. Closed versions of meiotic HORMADs assemble on meiotic chromosomes to drive pairing, synapsis, recombination, and meiotic progression [19]. While stable, open conformers of meiotic HORMADs have not been observed in vitro or in vivo, the budding yeast meiotic HORMAD, Hop1, exists in solution as two distinct conformations that are in equilibrium: the closed conformation, bound to a peptide fragment containing a closure motif from an interacting protein, and an intermediate "extended" state that has been referred to as unbuckled [28]. The physiological relevance of unbuckled meiotic HORMADs in vivo is unclear but work in plants and budding yeast indicates that their unbuckling is required for nuclear import and association with meiotic chromosomes [29,30].

While studies with other HORMADs, such as Mad2 and Rev7, provide a powerful framework to conceptualize PCH-2's molecular mechanism during meiotic prophase, identifying PCH-2's specific role during meiosis has been challenging. In contrast to Mad2 and Rev7, attempts to undertake similar structural and biochemical analysis of PCH-2/TRIP13 and meiotic HORMADs have been less informative due to technical difficulties. However, prior evidence from budding yeast and plants indicates that PCH-2/TRIP13 binds and acts directly on meiotic HORMADs. In budding yeast, Pch2 has been shown to bind to and remodel the single meiotic HORMAD, Hop1, in vitro [31] and in Arabidopsis, PCH2 immunoprecipitates the meiotic HORMAD, ASY1 [35]. Additionally, PCH-2's orthologs in other systems have been shown to promote meiotic progression [32] by depleting or redistributing meiotic HORMADs from chromosome axes upon synapsis, including budding yeast [33,34], Arabidopsis [35] and mice [10]. In mice, the removal of the meiotic HORMAD, HORMAD1, at the onset of synapsis is abolished in both TRIP13 mutants [10] or when the N-terminus of HORMAD1 is deleted [27], indicating that PCH-2 and its orthologs' remodeling of meiotic HORMADs is similar to its remodeling of Mad2 and Rev7. However, in some systems, such as C. elegans, meiotic HORMADs do not rely on PCH-2 for nuclear import [2] nor are they removed from chromosomes upon synapsis [21,22,36]. Further, PCH-2 and its orthologs localize to meiotic chromosomes as foci prior to synapsis in budding yeast, plants and C. elegans [2,34,35,37,38] and meiotic HORMADs are not fully removed during synapsis in budding yeast, plants or mice

[10,33,35]. These data raise the important question of whether remodeling of meiotic HORMADs plays additional roles during meiotic prophase.

Here, we use a combination of genetic, cytological and biochemical data to argue that PCH-2 remodels meiotic HORMADs to regulate and coordinate pairing, synapsis, crossover recombination, and meiotic progression in *C. elegans*. We show that a missense mutation in the HORMA domain of HTP-3 suppresses pairing and synapsis defects in *pch-2* mutants but exacerbates its recombination defects, indicating these two proteins cooperate to promote pairing and synapsis but regulate recombination independently. We also show that a similar mutation in *him-3* exacerbates synapsis defects in *pch-2* mutants but suppresses its recombination defects, suggesting that PCH-2 acts on HIM-3 specifically to promote crossover recombination. Finally, we show that corresponding mutations in the meiotic HORMAD HTP-1 produce a delay in meiotic progression, with consequences for synapsis and recombination, and can be partially suppressed by a mutation in *pch-2*, suggesting that PCH-2 cooperates with HTP-1 to regulate meiotic progression. Biochemical analysis demonstrates that these mutations limit the ability of meiotic HORMADs to adopt their closed conformations. Taken together, our work strongly supports a model in which PCH-2's remodeling of meiotic HORMADs from closed to unbuckled conformations decelerates pairing, synapsis, recombination, and meiotic progression, to proofread homolog interactions and maintain meiotic fidelity. Further, PCH-2 cleanly delegates its regulation of different meiotic prophase events to different meiotic HORMADs, offering new insight into how PCH-2 mechanistically coordinates the events of pairing, synapsis, and recombination. Lastly, it provides a potential explanation for why this family of meiotic HORMADs has dramatically expanded in *C. elegans* and provides a possible explanation about why some model systems have multiple meiotic HORMADs.

## Results

### PCH-2's localization is consistent with remodeling HORMADs on the axial elements of meiotic chromosomes

Meiotic HORMADs collaborate with cohesins and other axial proteins to organize meiotic chromosomes into linear axial elements functionally capable of undergoing interhomolog pairing, synapsis and recombination [5]. While PCH-2/TRIP13 has been implicated in remodeling meiotic HORMADs away from meiotic chromosomes to ensure their availability [29,39] and entry into the nucleus [30], its localization to meiotic chromosomes before and after synapsis [2,34,35,37,38] suggests a role remodeling meiotic HORMADs on chromosomes to regulate pairing, synapsis and recombination as well as meiotic progression. To test this, we used structured illumination microscopy to more closely investigate whether PCH-2's localization was consistent with remodeling meiotic HORMADs on axial elements.

Before synapsis and during axis assembly, when meiotic HORMADs are diffuse throughout nuclei, PCH-2 localizes to foci on meiotic chromosomes [2]. After synapsis, PCH-2 localizes along the synaptonemal complex (SC) and this localization depends on the central element component SYP-1 [2,40], which assembles between axial elements during synapsis (Fig 1A). We performed indirect immunofluorescence on wildtype meiotic nuclei against both PCH-2 and SYP-1 and performed structured illumination microscopy to resolve the distance between axial elements (Fig 1B) [41]. While SYP-1 appeared as a line of continuous width in all meiotic nuclei, PCH-2 staining often appeared wider than SYP-1, sometimes splitting as parallel tracks with SYP-1 staining between PCH-2 tracks (inset and arrows in Fig 1B and 1C). Indeed, when we quantified the signal intensities for both PCH-2 and SYP-1, we found that the signal for PCH-2 spanned a wider distance compared to SYP-1 (Fig 1D), indicating that PCH-2 can

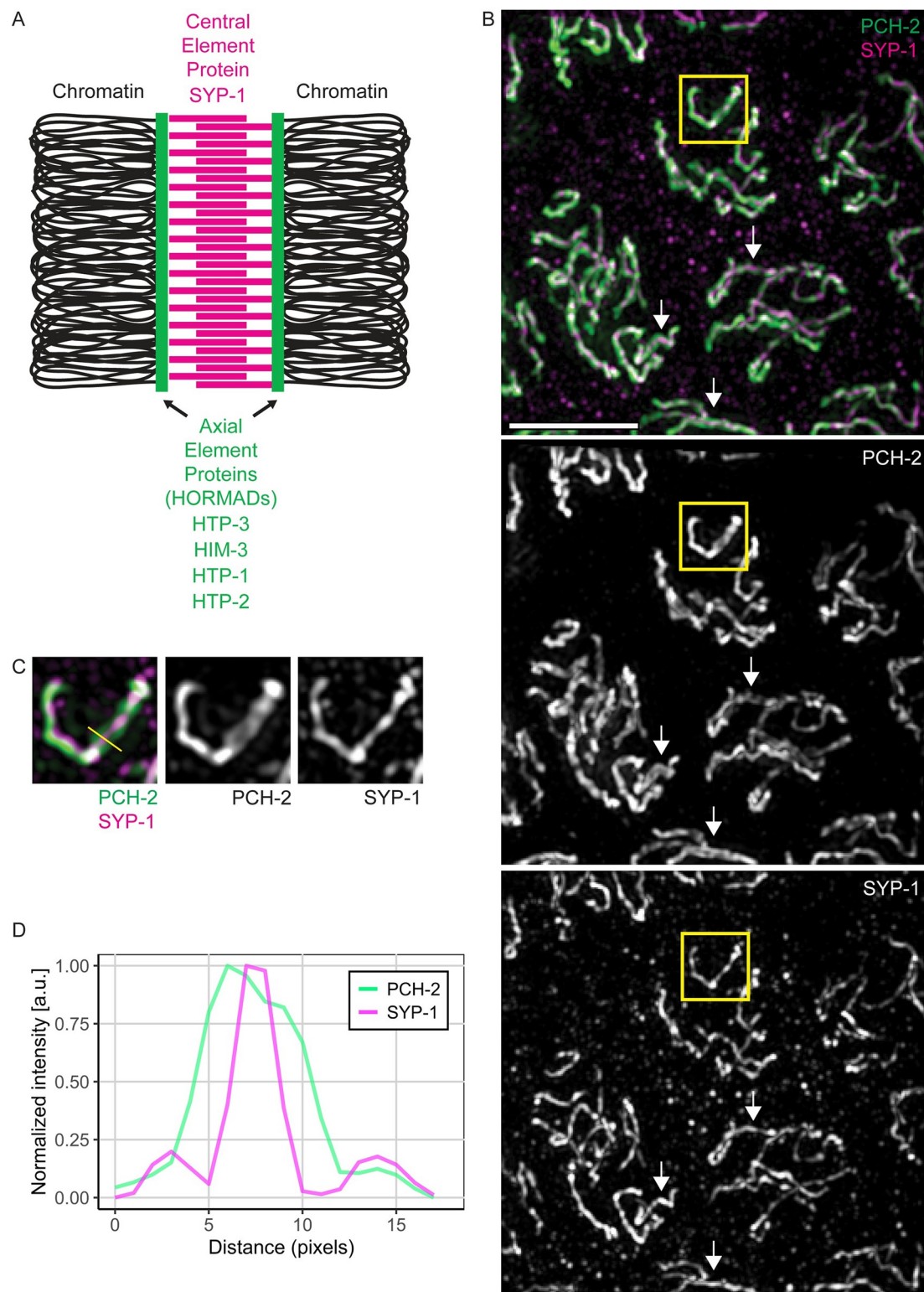

**Fig 1. PCH-2's localization is consistent with remodeling HORMADs on meiotic axes. A.** Cartoon of synaptonemal complex with axial elements made up of meiotic HORMADs and the central element component, SYP-1. **B.** Structured illumination microscopy (SIM) images of meiotic nuclei stained with antibodies against PCH-2 (green) and SYP-1 (magenta). Arrows represent regions of PCH-2 potentially associated with meiotic axes. Scalebar indicates 5 microns. **C.** Cropped insets of PCH-2 and SYP-1 signal. Yellow line represents area of pixel intensities measured. **D**. Line scans of PCH-2 (green) and SYP-1 (magenta) pixel intensities plotted against distance in pixels.

reside on or close to axial elements of synapsed chromosomes and potentially remodel meiotic HORMADs on chromosomes.

## *htp-3*$^{H96Y}$ suppresses the acceleration of pairing and synapsis in *pch-2* mutants

Having established that PCH-2's localization is consistent with the ability to remodel meiotic HORMADs on chromosomes, we looked for genetic interactions that might support this possibility. *htp-3(vc75)* is a missense mutation that replaces histidine 96 with a tyrosine in the HORMA domain (H96Y) (Fig 2A). HTP-3$^{H96Y}$ properly localizes to meiotic chromosomes [42] and does not affect pairing, synapsis and crossover recombination [42,43]. However, we and others previously showed that this mutation in *htp-3* abolished meiotic checkpoint responses in *C. elegans* [42,43]. Because of this phenotype, we analyzed HTP-3's relationship with PCH-2 in regulating pairing, synapsis, recombination, and meiotic progression.

To test whether PCH-2 genetically interacts with HTP-3 to regulate pairing, we generated *pch-2;syp-1* double mutants, *htp-3*$^{H96Y}$*;syp-1* double mutants, and *pch-2;htp-3*$^{H96Y}$*;syp-1* triple mutants and assessed pairing when compared with *syp-1* single mutants. *syp-1* null mutants successfully undergo pairing despite the failure to load SC components, particularly at specific loci called Pairing Centers (PCs) [40]. Therefore, these experiments allow us to analyze pairing independent of synapsis. We performed immunofluorescence against the HIM-8 protein, which localizes to PCs on the X chromosome [44], to assess pairing at this locus throughout the germline: two HIM-8 foci indicate the X-chromosomes are unpaired while a single HIM-8 focus indicates paired X-chromosomes. Because meiotic nuclei are arranged in spatiotemporal gradient in germlines, we divided the germline into six equivalently sized zones to perform a meiotic time-course and determined the percentage of nuclei with a single HIM-8 focus in each zone (Fig 3A and 3B).

Similar to previous work [2,3], we found that there is an increase in percentage of nuclei with paired X chromosomes in zone 2 for *pch-2;syp-1* double mutants (76%, Fig 3C) compared to *syp-1* single mutant controls (49%, Fig 3C), showing that the rate of pairing is accelerated in the absence of *pch-2* (p value <0.0001 by two-tailed Fisher's exact test). *htp-3*$^{H96Y}$*;syp-1* double mutants have a similar rate of pairing compared to *syp-1* single mutant controls (46%, Fig 3C, p value = 0.4001 by two-tailed Fisher's exact test), demonstrating that this mutation does not affect the rate of pairing. Interestingly, we found that *pch-2;htp-3*$^{H96Y}$*;syp-1* triple mutants have a similar rate of pairing compared to *syp-1* single mutants and *htp-3*$^{H96Y}$*;syp-1* double mutants (49%, Fig 3C), indicating that *htp-3*$^{H96Y}$ suppresses the acceleration of pairing observed in *pch-2* mutants.

Next, we tested for a genetic interaction between *pch-2* and *htp-3* during synapsis. We analyzed the rate of synapsis in germlines from control, *pch-2* single mutants, *htp-3*$^{H96Y}$ single mutants, and *pch-2;htp-3*$^{H96Y}$ double mutant worms by performing immunofluorescence against the axial element HTP-3 [12,36] and the central element SYP-1 [40]. We then determined the fraction of nuclei with complete synapsis in each zone. Complete synapsis is visible as full co-localization of HTP-3 and SYP-1, whereas nuclei with incomplete synapsis have stretches of HTP-3 without SYP-1 signal (arrowheads, Fig 3D). Similar to previous data, we report that *pch-2* null mutants have an accelerated rate of synapsis (61%, Fig 3E) compared to wildtype controls (32%, Fig 3E, p value <0.0001 by two-tailed Fisher's exact test) in zone 2 and a delay in desynapsis in zone 6 [2,3] (Fig 3E). *htp-3*$^{H96Y}$ single mutants have a similar rate of synapsis compared to wildtype controls (36%, Fig 3E, p value = 0.228 by two-tailed Fisher's exact test), suggesting that the rate of synapsis is unaffected in this background, and accelerated desynapsis, as previously reported [42]. Analogous to our pairing results, we found that

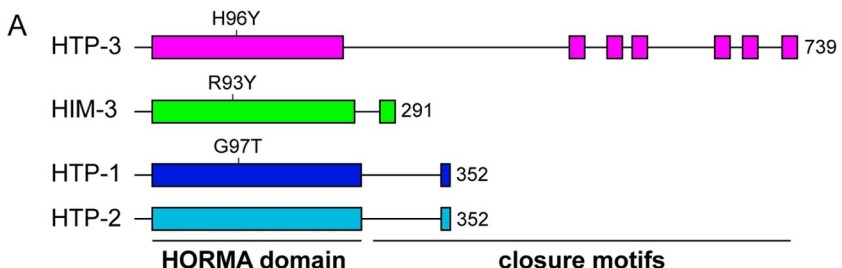

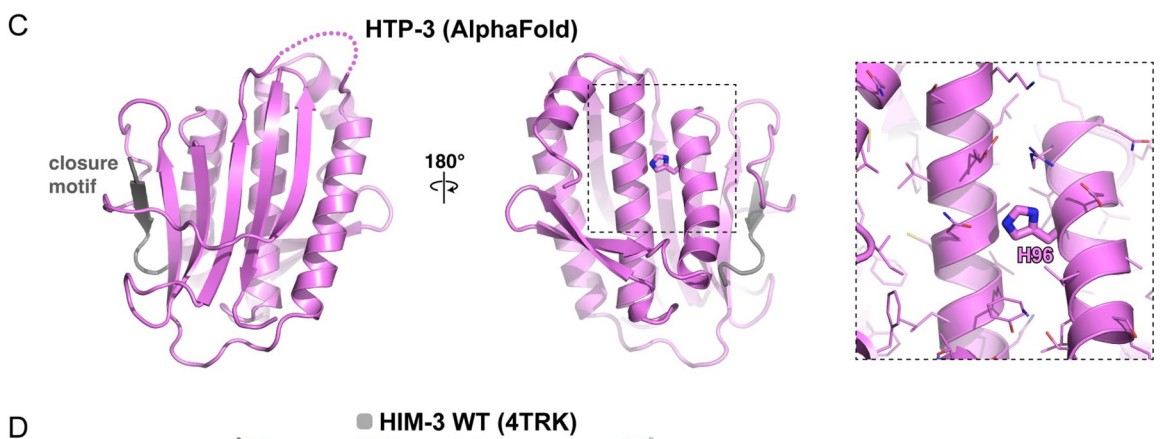

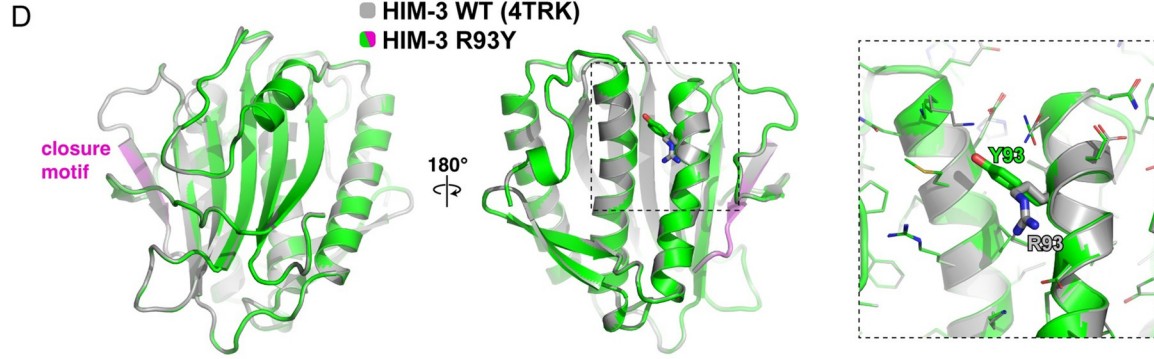

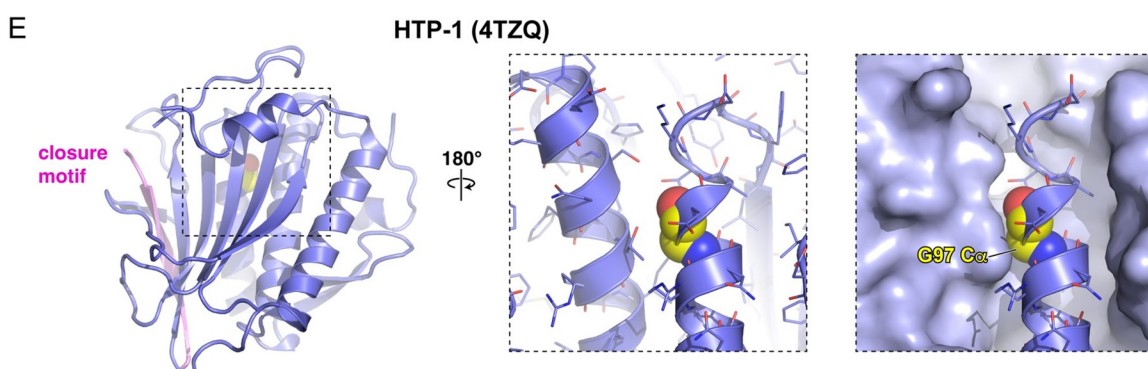

**Fig 2. HORMA domain structure of meiotic HORMADs in *C. elegans*. A.** Cartoon depicting protein domains of HTP-3, HIM-3, HTP-1, and HTP-2. HORMA domains and closure motifs are shown as colored boxes and labeled. Mutations generated by CRISPR/ Cas9 are shown. **B.** Clustal Omega alignment of the HORMA domains of HTP-3, HIM-3, HTP-1, and HTP-2. Asterisks represent conserved residues,: represents conservative changes and. represents semi-conservative changes. Red box indicates targeted reside for mutation for each respective HORMAD. **C.** Structure of the *C. elegans* HTP-3 HORMA domain (pink) bound to a closure motif (gray), predicted by AlphaFold 2 [93]. Right panels show the position of H96. **D.** Overlay of wild-type (gray; PDB ID 4TRK) and R93Y mutant (green/pink) of *C. elegans* HIM-3. Right panels show the position of residue 93 and show that the overall HORMA domain structure is not impacted upon mutation of arginine 93 to tyrosine. **E.** Structure of the *C. elegans* HTP-1 HORMA domain (PDB ID 4TZQ). Right panels show the position of residue G97 (yellow spheres), which is partially buried against a neighboring alpha-helix.

*pch-2;htp-3$^{H96Y}$* double mutants have a similar rate of synapsis compared to wildtype controls and *htp-3$^{H96Y}$* single mutants (34%, Fig 3E), indicating that *htp-3$^{H96Y}$* suppresses the acceleration of synapsis in *pch-2* mutants.

To further solidify that PCH-2 interacts with HTP-3 to regulate synapsis, we analyzed synapsis in *meDf2* heterozygotes (*meDf2/+*). *meDf2* is a mutation in which PC regions are deleted from the X chromosome [45]. Since PCs are essential for pairing and synapsis [36], *meDf2* homozygotes fail to pair and synapse X chromosomes while *meDf2* heterozygotes behave like a partial loss of function mutation: ~50% of nuclei complete synapsis of X chromosomes while the remaining 50% have unsynapsed X chromosomes [36]. In *pch-2;meDf2/+* worms, ~85% of nuclei complete synapsis, demonstrating that PCH-2 normally inhibits synapsis when PC function is compromised [2].

To test whether *htp-3$^{H96Y}$* also suppressed the synapsis in *pch-2;meDf2/+* worms, we generated *pch-2;htp-3$^{H96Y}$;meDf2/+* triple mutants and analyzed the rate of synapsis, comparing it to controls. Similar to previous results, we found that 87% of nuclei in *pch-2;meDf2/+* double mutants achieve complete synapsis in zone 5 compared to 48% complete synapsis for *meDf2/+* single mutant worms (Fig 3F). *htp-3$^{H96Y}$;meDf2/+* worms have 52% of nuclei with complete synapsis in zone 5, similar to *meDf2/+* single mutants (Fig 3F). *pch-2;htp-3$^{H96Y}$;meDf2/+* triple mutants show maximal levels of synapsis in zone 4 (36%, Fig 3F) and more closely resemble the levels of synapsis observed in to *htp-3$^{H96Y}$;meDf2/+* double and *meDf2/+* single mutants than *pch-2;meDf2/+* double mutants. These data indicate that PCH-2's ability to inhibit synapsis when PC function is compromised depends on the function of HTP-3, and perhaps more specifically, on the HORMA domain of HTP-3, similar to what we observed when PCs are fully functional (Fig 3E). To determine whether these defects in synapsis produce chromosome segregation errors, we scored the percent of male progeny in these strains. Male progeny, which are genotypically XO in contrast to XX hermaphrodites, are diagnostic of misegregation of X chromosomes in meiosis. The ability of *htp-3$^{H96Y}$* to suppress the synapsis of X chromosomes in *pch-2;meDf2/+* was further supported by the increase in male progeny observed in *pch-2;htp-3$^{H96Y}$;meDf2/+* triple mutants (10.84%, Table 1) in comparison to *pch-2;meDf2/+* double mutants (0.83%, Table 1).

## *htp-3$^{H96Y}$* exacerbates defects in DNA repair and crossover formation in *pch-2* mutants

We next tested whether PCH-2 regulates recombination through HTP-3 by assaying DNA repair and crossover formation in *pch-2;htp-3$^{H96Y}$* double mutants. The rate of DNA repair is assessed by quantifying the number of RAD-51 foci per nucleus throughout the germline (Fig 3G). The appearance of RAD-51 foci on meiotic chromosomes is associated with the introduction of DNA double-strand breaks and their disappearance is associated with the entry of DNA double-strand breaks into repair pathways [46]. In stark contrast to what has been observed in yeast, mice and plants, *C, elegans* have substantially fewer RAD-51 foci and the number of these foci peak later in meiotic prophase [46]. For example, RAD-51 numbers

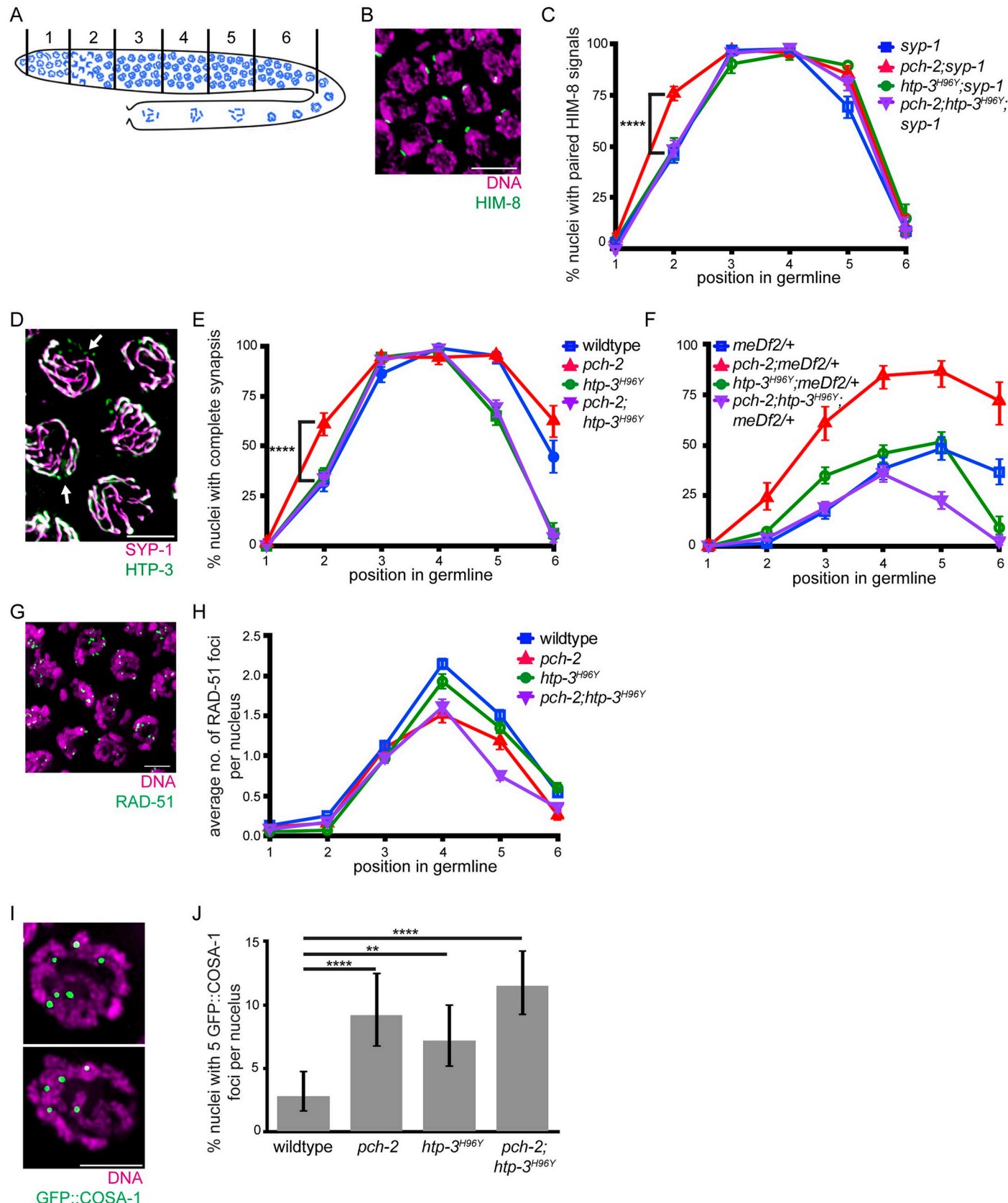

**Fig 3. *htp-3*[H96Y] suppresses the acceleration of pairing and synapsis but exacerbates defects in DNA repair and crossover formation in *pch-2* mutants. A.**
Cartoon depicting zones for quantification in the *C. elegans* germline. **B.** Meiotic nuclei in zone 2 in *syp-1* mutant worms stained with DAPI (magenta) and antibody against HIM-8 (green). **C.** Quantification of percent nuclei with paired HIM-8 signals in zones 1–6 for *syp-1* (blue), *pch-2;syp-1* (red), *htp-3*[H96Y];*syp-1* (green) and *pch-2;htp-3*[H96Y];*syp*-1 (purple) mutant strains. Error bars represent 95% confidence intervals. **D.** Meiotic nuclei in zone 3 in wildtype worms stained with antibodies against SYP-1(magenta) and HTP-3 (green). Arrows indicates unsynapsed chromosomes. **E.** Quantification of percent nuclei with complete synapsis in zones 1–6 for wildtype (blue), *pch-2* (red), *htp-3*[H96Y] (green) and *pch-2;htp-3*[H96Y] (purple) mutant strains. Error bars represent 95% confidence intervals. **F.** Quantification of percent nuclei with complete synapsis in zones 1–6 for *meDf2/+* (blue), *pch-2;meDf2/+* (red), *htp-3*[H96Y];*meDf2/+* (green) and *pch-2;htp-3*[H96Y];*meDf2/+* (purple) mutant strains. Error bars represent 95% confidence intervals. **G.** Meiotic nuclei in zone 4 in wildtype worm stained with DAPI (magenta) and antibody against RAD-51 (green). **H.** Quantification of average number of RAD-51 foci in zones 1–6 for wildtype (blue), *pch-2* (red), *htp-3*[H96Y] (green) and *pch-2;htp-3*[H96Y] (purple) mutant strains. Error bars represent standard error of the mean (SEM). **I.** Meiotic nuclei from control and *pch-2* mutant worms stained with DNA (magenta) and antibody against GFP::COSA-1 (green). Top indicates representative nuclei with 6 GFP::COSA-1 foci. Bottom indicates representative nuclei with 5 GFP::COSA-1 foci. **J.** Quantification of percent nuclei with 5 GFP::COSA-1 foci for wildtype (n = 464), *pch-2* (n = 382), *htp-3*[H96Y] (n = 433), and *pch-2;htp-3*[H96Y] (n = 602) mutant strains. Error bars represent 95% confidence intervals. In all images, scalebar indicates 5 microns. In all graphs, ** indicates p-value < 0.01 and **** indicated p-value <0.0001, Significance was assessed using two-tailed Fisher's exact-tests.

peak in zone 4 in wildtype animals, increasing to an average of 2 per nuclei, and decrease as RAD-51 is removed during DNA repair. *pch-2* single mutants have a lower average of RAD-51 foci per nucleus in zones 4–6 (Fig 3H), consistent with previous findings that that *pch-2* mutants exhibit a faster rate of DNA repair compared to wildtype controls [2]. *htp-3*[H96Y] single mutants show a similar rate of DNA repair as wildtype germlines (Fig 3H), although the average number of RAD-51 foci is slightly lower than wildtype [42]. By contrast, *pch-2;htp-3*[H96Y] double mutants not only show an accelerated rate of DNA repair, similar to *pch-2* single mutants, they have a reduced average number of RAD-51 foci in zone 5 (0.75 foci, Fig 3H), showing that mutation of *htp-3* exacerbates the acceleration of DNA repair in *pch-2* mutants. Because *htp-3*[H96Y] and *pch-2* mutants do not have reduced numbers of DNA double-stranded breaks [2,42], these results reflect changes in the rate of DNA repair. Further, this indicates that, in contrast to our findings for pairing and synapsis, PCH-2 does not act through HTP-3 to regulate the rate of DNA repair. Instead, both proteins regulate DNA repair independent of each other.

We previously reported that *pch-2;htp-3*[H96Y] double mutants exhibit more achiasmate chromosomes than either single mutant [2], indicating that the double mutant has a more severe defect in meiotic crossover recombination than either single mutant and consistent with our analysis of DNA repair (Fig 3H). To further verify this result on crossover formation, we quantified the number of GFP::COSA-1 foci for each meiotic nucleus. COSA-1 is a cyclin-like

**Table 1. Viability and percent male-self-progeny.** Table with percent viable progeny and males. Total number of progeny listed in parentheses.

| genotype | percent viability (total no. of animals) | percent male self-progeny |
|---|---|---|
| wildtype | 100% (2489) | 0.22% |
| *meDf2/+* | 97% (2286) | 10.91% |
| *pch-2* | 100% (2388) | 0.71% |
| *pch-2;meDf2/+* | 98.15% (1808) | 0.83% |
| *htp-3*[H96Y] | 100% (2586) | 0.12% |
| *htp-3*[H96Y];*meDf2/+* | 97.03% (1957) | 11.34% |
| *pch-2;htp-3*[H96Y] | 81.89% (1067) | 1.12% |
| *pch-2;htp-3*[H96Y];*meDf2/+* | 83.90% (1652) | 10.84% |
| *him-3*[R93Y] | 100% (3304) | 0.15% |
| *him-3*[R93Y];*meDf2/+* | 97.69% (3170) | 11.30% |
| *pch-2;him-3*[R93Y] | 100% (3027) | 0.03% |
| *pch-2;him-3*[R93Y];*meDf2/+* | 98.43% (3140) | 10.34% |
| *htp-1*[G97T] | 67.29% (1984) | 2.25% |
| *pch-2;htp-1*[G97T] | 58.44% (2175) | 1.97% |

protein that localizes specifically to presumptive crossovers, serving as a cytological marker for crossover formation [47]. Because *C. elegans* contain 6 pairs of chromosomes and exhibit strong crossover interference, most wildtype nuclei have 6 GFP::COSA-1 foci per nucleus (Fig 3I) [47]. Our findings also reflect this, where we report only 3% of nuclei have less than 6 GFP::COSA-1 foci (Fig 3J). We also found that the incidence of nuclei with less than 6 GFP:: COSA-1 foci increases to 9% of nuclei in *pch-2* single mutants (Fig 3J, p value < 0.0001 by Fisher's exact test), recapitulating previous findings that *pch-2* mutants exhibit a loss of cross-over assurance [2,3]. *htp-3*$^{H96Y}$ single mutants also show an increase in the number of nuclei with less than 6 GFP::COSA-1 crossovers compared to wildtype (7%, Fig 3J, p value = 0.003 by Fisher's exact test), consistent with their checkpoint role [42]. Similar to our analysis of DNA repair, the number of nuclei with less than 6 GFP::COSA-1 foci in *pch-2;htp-3*$^{H96Y}$ double mutants increases to 11% percent (Fig 3J, p value <0.0001 by Fisher's exact test), showing an additive effect for crossover defects. These data illustrate that PCH-2 and HTP-3 work independently to regulate crossover formation.

Altogether, our analysis of pairing, synapsis and crossover recombination indicate that PCH-2 and HTP-3 act together to regulate pairing and synapsis but independently to regulate DNA repair and crossover formation (Table 2). Since *htp-3*$^{H96Y}$ single mutants also show defects in the DNA damage response [43] while *pch-2* mutants do not [48] (see Table 2), this separation of function is not necessarily surprising. However, more importantly these data strongly support a model in which PCH-2 acts through HTP-3's HORMA domain to control the rate of pairing and synapsis but not the process of meiotic recombination.

## *htp-3*$^{H96Y}$ does not affect meiotic progression

In budding yeast, Pch2's works with the meiotic HORMAD, Hop1, to control meiotic progression [32]. Since *htp-3*$^{H96Y}$ mutants genetically interact with *pch-2* mutants in *C. elegans*, we also tested whether *htp-3*$^{H96Y}$ mutants affected meiotic progression. We previously reported that *pch-2* mutants do not exhibit defects in meiotic progression despite the acceleration of pairing, synapsis and DNA repair in these mutants [2]. We assessed meiotic progression by performing immunofluorescence against a factor required for DNA double-strand break formation, DSB-1 (S1A Fig). DSB-1 localizes to meiotic chromosomes to facilitate DNA double-strand break formation [49]. In response to meiotic defects, DSB-1 persists on chromosomes to maintain a period of competency for DNA double-strand break formation to promote crossover assurance [49] and this persistence correlates with other markers of delays in meiotic progression [50,51]. We quantified the fraction of meiotic nuclei that were positive for DSB-1 and could not detect any difference between control and *htp-3*$^{H96Y}$ mutant germlines (S1B Fig, p value = 0.8269 by Fisher's exact test), illustrating that this mutation does not affect meiotic progression.

A null mutation in *htp-3* abolishes the delay in meiotic progression observed when there are defects in synapsis, such as in *syp-1* mutants [50]. To determine whether *htp-3*$^{H96Y}$ affected meiotic progression in *syp-1* mutants, we quantified the fraction of DSB-1 positive meiotic

**Table 2. Summary of phenotypes associated with meiotic HORMAD mutations described in this study.**

| | *htp-3*$^{H96Y}$ | *him-3*$^{R93Y}$ | *htp-1*$^{G97T}$ |
|---|---|---|---|
| **PCH-2 independent** | • defective in DNA damage checkpoint [42,43]<br>• accelerated desynapsis [42]<br>• promotes crossover assurance (this study) | • regulates synapsis (this study) | • regulates synapsis (this study)<br>• regulates crossover control (this study) |
| **PCH-2 dependent** | • regulates pairing (this study)<br>• regulates synapsis (this study) | • promotes crossover assurance (this study) | • regulates meiotic progression (this study) |

nuclei in both *syp-1* single mutants and *syp-1;htp-3$^{H96Y}$* double mutants and could not observe any difference between these genotypes (S1C Fig, p value = 1.0 by Fisher's exact test), indicating that *htp-3$^{H96Y}$* does not affect the extension of meiotic progression observed when there are meiotic defects.

## *him-3$^{R93Y}$* does not suppress the acceleration of pairing or synapsis in *pch-2* mutants

Our findings that PCH-2 acts through HTP-3 to promote pairing and synapsis, but not recombination, indicates that PCH-2 regulates recombination through an HTP-3 independent mechanism. Further, it is unclear whether PCH-2 regulates pairing and synapsis only through its interaction with HTP-3 or whether other meiotic HORMADs also contribute to this regulation. To address these possibilities, we next tested whether *pch-2* genetically interacts with similar mutations in *him-3* or *htp-1*, genes that encode the two other HORMADs essential for meiosis in *C. elegans*.

We performed a Clustal Omega alignment of HTP-3 with the other three meiotic HORMADs [52] and observed that the mutation in *htp-3$^{H96Y}$* was not of a conserved residue but was adjacent to two invariant residues in all four meiotic HORMADs (red box in Fig 2B). Further, based on this alignment, the arginine at position 93 of HIM-3 should be positioned similarly within the HORMA domain (Fig 2C and 2D). Using CRISPR/Cas9 gene editing, we made a corresponding missense mutation to the *htp-3$^{H96Y}$* allele in *him-3*, where arginine 93 was replaced with a tyrosine (*him-3$^{R93Y}$*). Indirect immunofluorescence verified that this mutation did not affect HIM-3's localization to chromosomes (S2 Fig), progeny viability or the production of male progeny (Table 1).

First we analyzed pairing in *syp-1*, *pch-2;syp-1*, *him-3$^{R93Y}$;syp-1*, and *pch-2;him-3$^{R93Y}$;syp-1* mutants by performing immunofluorescence against HIM-8 (Fig 4A and 4B). *him-3$^{R93Y}$;syp-1* double mutants show 44% of nuclei completed pairing in zone 2, similar to *syp-1* single mutants (p value = 1.00 by two-tailed Fisher's exact test), demonstrating that this mutation does not affect pairing (Fig 4B). In contrast to our analysis of the *htp-3$^{H96Y}$* allele, we found that *pch-2;him-3$^{R93Y}$;syp-1* triple mutants also show an accelerated rate of pairing phenotype (63%, Fig 4B) similar to that of *pch-2;syp-1* double mutants (p value = 0.52 by two-tailed Fisher's exact test). Therefore, we detect no genetic interaction between *pch-2* and *him-3$^{R93Y}$* in affecting the rate of pairing.

Next, we assessed the rate of synapsis in wildtype, *pch-2*, *him-3$^{R93Y}$*, and *pch-2;him-3$^{R93Y}$* germlines by performing immunofluorescence against SYP-1 and HTP-3 (Fig 4C and 4D). In both *pch-2* and *him-3$^{R93Y}$* single mutants, we found that there was an increase in the number of nuclei that completed synapsis in zone 2 (13% for both, compared to the wildtype control of 3%, Fig 4D), suggesting that synapsis is accelerated in both mutants (p values <0.0001 for both comparisons by two-tailed Fisher's exact test). However, we found that *pch-2;him-3$^{R93Y}$* double mutants display more severe synapsis defects than either single mutant: the rate of synapsis is delayed throughout zones 2–5 and these double mutants reach only 87% complete synapsis throughout the germline (Fig 4D). These results demonstrate that PCH-2 and HIM-3 regulate the progression of synapsis independent of each other.

Finally, we further assessed synapsis in *him-3$^{R93Y}$* and *pch-2;him-3$^{R93Y}$* mutants by scoring synapsis in *meDf2* heterozygotes (*meDf2/+*). Consistent with a role in regulating synapsis, *him-3$^{R93Y}$;meDf2/+* double mutants show a reduced rate of synapsis (Fig 4E) compared to *meDf2/+* controls (see zones 4 and 5). This effect was intermediate in *pch-2;him-3$^{R93Y}$;meDf2/+* triple mutant worms, providing further support that PCH-2 and HIM-3 both regulate synapsis but accomplish this through independent mechanisms.

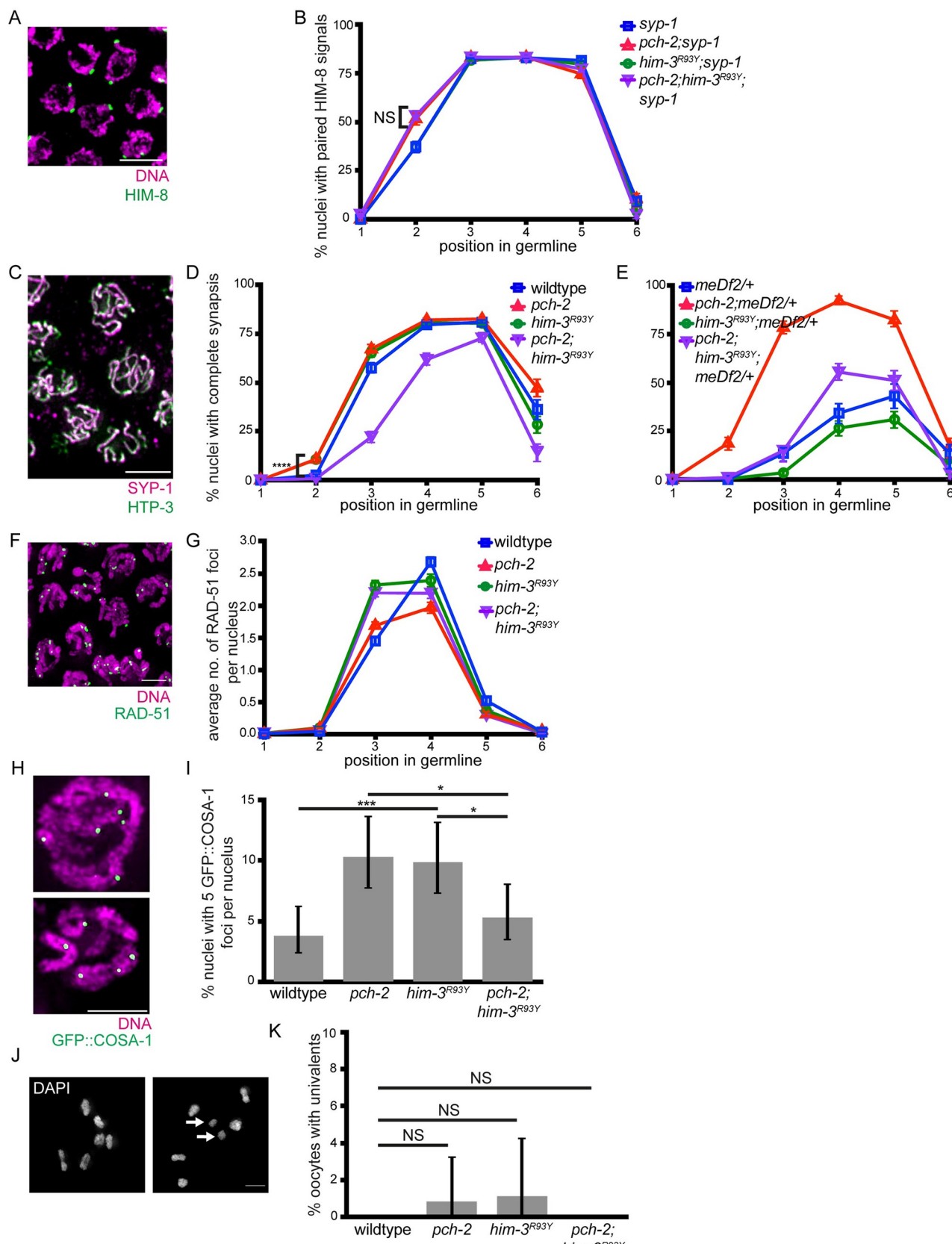

**Fig 4. *him-3$^{R93Y}$* does not suppress the acceleration of pairing, synapsis, or DNA repair, but does suppress defects in crossover formation in *pch-2* mutants. A.** Meiotic nuclei in zone 2 in *syp-1* mutants stained with DAPI (magenta) and antibody against HIM-8 (green). **B.** Quantification of percent nuclei with paired HIM-8 signals in zones 1–6 for *syp-1* (blue), *pch-2;syp-1* (red), *him-3$^{R93Y}$;syp-1* (green) and *pch-2;him-3$^{R93Y}$;syp-1* (purple) mutant strains. Error bars represent 95% confidence intervals. **C.** Meiotic nuclei in zone 3 in wildtype worms stained with antibodies against SYP-1 (magenta) and HTP-3 (green). Arrow indicates unsynapsed chromosomes. **D.** Quantification of percent nuclei with complete synapsis in zones 1–6 for wildtype (blue), *pch-2* (red), *him-3$^{R93Y}$* (green) and *pch-2;him-3$^{R93Y}$* (purple) mutant strains. Error bars represent 95% confidence intervals. **E.** Quantification of percent nuclei with complete synapsis in zones 1–6 for *meDf2/+* (blue), *pch-2;meDf2/+* (red), *him-3$^{R93Y}$;meDf2/+* (green) and *pch-2; him-3$^{R93Y}$;meDf2/+* (purple) mutant strains. Error bars represent 95% confidence intervals. **F.** Meiotic nuclei in zone 4 in wildtype worms stained with DAPI (magenta) and antibody against RAD-51 (green). **G.** Quantification of average number of RAD-51 foci in zones 1–6 for wildtype (blue), *pch-2* (red), *him-3$^{R93Y}$* (green) and *pch-2;him-3$^{R93Y}$* (purple) mutant strains. Error bars represent standard error of the mean (SEM). **H.** Meiotic nuclei from control and *pch-2* mutant worms stained with DNA (magenta) and antibody against GFP::COSA-1 (green). Top indicates representative nuclei with 6 GFP::COSA-1 foci. Bottom indicates representative nuclei with 5 GFP::COSA-1 foci. **I.** Quantification of percent nuclei with 5 GFP::COSA-1 foci for wildtype (n = 392), *pch-2* (n = 398), *him-3$^{R93Y}$* (n = 382), and *pch-2;him-3$^{R93Y}$* (n = 378) mutant strains. Error bars represent 95% confidence intervals. **J.** Oocytes from wildtype and *pch-2* mutant worms stained with DAPI. Left indicates representative oocyte with zero univalents. Right indicates representative oocyte with univalents (arrows). Scalebar indicates 5 microns. **K.** Quantification of percent oocytes with univalents for wildtype (n = 155), *pch-2* (n = 232), *him-3$^{R93Y}$* (n = 174), and *pch-2;him-3$^{R93Y}$* (n = 158) mutant strains. Error bars represent 95% confidence intervals. In all graphs, NS indicates non-significant, * indicates p-value < 0.05, ** indicates p-value < 0.01, *** indicates p-value < 0.001, and **** indicated p-value <0.0001. Significance was assessed using two-tailed Fisher's exact-tests. In all images, scalebar indicates 5 microns.

## *him-3$^{R93Y}$* does not suppress DNA repair defects in *pch-2* mutants

We next tested whether *pch-2* genetically interacts with *him-3$^{R93Y}$* during the regulation of DNA repair by scoring RAD-51 foci in N2, *pch-2*, *him-3$^{R93Y}$* and *pch-2;him-3$^{R93Y}$* mutants (Fig 4F). In *him-3$^{R93Y}$* single mutants, the average number of RAD-51 foci was higher in zone 3 than both *pch-2* and control germlines, suggesting a more rapid installation of RAD-51 and potentially more DNA double-strand breaks. However, the rate of DNA repair was similar to wildtype (Fig 4G). *pch-2;him-3$^{R93Y}$* worms more closely resembled *him-3$^{R93Y}$* single mutants than *pch-2* single mutants in zone 3 but were intermediate between *him-3$^{R93Y}$* and *pch-2* single mutants in zone 4 (Fig 4G), indicating that the accelerated rate of DNA repair observed in *pch-2* mutants is not fully suppressed by the *him-3$^{R93Y}$* mutation and that these two factors do not act in the same pathway to regulate DNA repair.

## *him-3$^{R93Y}$* suppresses crossover recombination defects in *pch-2* mutants

To analyze the effect of *him-3$^{R93Y}$* on crossover recombination, and test for a genetic interaction with *pch-2* mutants, we scored GFP::COSA-1 foci in wildtype, *pch-2*, *him-3$^{R93Y}$* and *pch-2; him-3$^{R93Y}$* mutant strains (Fig 4H). Similar to *pch-2* single mutants, we found that there was an increase of nuclei with 5 GFP::COSA-1 foci in *him-3$^{R93Y}$* mutants compared to wildtype controls (10%, Fig 4I, p value = 0.0004 by two-tailed Fisher's exact test), showing that mutation of this residue in HIM-3 causes a loss of crossover assurance. In contrast to *pch-2* mutants and *him-3$^{R93Y}$* single mutants, *pch-2;him-3$^{R93Y}$* double mutants have a reduced number of nuclei with less than 6 foci compared to either single mutant and more closely resembles the frequency observed in wildtype nuclei (5%, Fig 4I, p values = 0.016 and 0.0214 by two-tailed Fisher's exact test, respectively). Therefore, *him-3$^{R93Y}$* suppresses the defect in crossover assurance observed in *pch-2* mutants.

We next visualized whether we could observe this genetic suppression later in meiotic prophase, when chromosome pairs are linked by chiasmata. Wildtype *C. elegans* oocytes contain 6 pairs of chromosomes or bivalents, and these can be visualized as 6 "DAPI stained bodies." Defects in recombination produce homologs without chiasmata, or univalents, in oocytes (arrows, Fig 4J). In our analysis of wildtype oocytes, we did not observe any univalents (Fig 4K). Similar to our previous findings [2], we found that there was an increase in the percent of nuclei with univalents in *pch-2* single mutants compared to wildtype controls (0.8%, Fig 4K), further demonstrating a loss of crossover assurance that leads to achiasmate homolog pairs. While this difference is not statistically significant, it is biologically significant, given that we

never see univalents in wildtype oocytes. Correlating with our GFP::COSA-1 analysis, *him-3$^{R93Y}$* single mutants also have an increase in the number of oocytes containing univalents (1.3%, Fig 4K), demonstrating that *him-3$^{R93Y}$* mutants also observe a loss of crossover assurance. In *pch-2;him-3$^{R93Y}$* double mutants, we found no instances of univalents (Fig 4K), further indicating that these two mutant alleles genetically suppress each other in the context of crossover recombination.

Taken together, our results are consistent with a model in which PCH-2 acts through HIM-3 to promote crossover recombination but not pairing or synapsis (Table 2). Further, the roles of the two meiotic HORMADs appear to be cleanly delegated, with PCH-2 controlling pairing and synapsis through HTP-3 and crossover recombination through HIM-3. Since we do not observe a similar genetic interaction when we analyze DNA repair in *pch-2;him-3$^{R93Y}$* double mutants, this regulation of crossover recombination occurs later than the installation and removal of RAD-51.

### *him-3$^{R93Y}$* does not affect meiotic progression

We next tested whether *him-3$^{R93Y}$* mutants affect meiotic progression by performing immunofluorescence against DSB-1 and quantifying the fraction that were DSB-1 positive in wildtype and *him-3$^{R93Y}$*mutant strains (S3A Fig). We found that there was not a significant difference between the number of DSB-1 positive nuclei between wildtype and *him-3$^{R93Y}$* mutant strains (S3B Fig, p value = 0.645 blank by two-tailed Fisher's exact test), suggesting that this mutation does not affect meiotic progression.

We further tested whether the *him-3$^{R93Y}$* mutation affected meiotic progression by quantifying the fraction of DSB-1 positive nuclei in *syp-1* and *him-3$^{R93Y}$;syp-1* mutant strains. Similar to our findings for *htp-3$^{H96Y}$*, we found that there was not a significant difference in the number of DSB-1 positive nuclei between *syp-1* single mutant*s* and *him-3$^{R93Y}$;syp-1* double mutants (S3C Fig, p value = 1.00 by two-tailed Fisher's exact test). This result further demonstrates that *him-3$^{R93Y}$* does not affect meiotic progression.

### *htp-1$^{G97T}$* does not suppress the accelerated rate of pairing in *pch-2* mutants

Our findings suggest that PCH-2 acts through HTP-3 to promote pairing and synapsis, and HIM-3 to promote crossover formation. We next wanted to test whether *pch-2* genetically interacts with a similar mutation in *htp-1*, the gene that encodes the final essential meiotic HORMAD in *C. elegans*.

Preliminary structural analysis of the HORMA domain of HTP-1 indicates that the glycine residue that is analogous to the histidine and arginine residues of HTP-3 and HIM-3 (Fig 2B), respectively, is positioned more internally within the HORMA domain (Fig 2E) and therefore potentially more constrained in the mutations we can introduce at this site without affecting HTP-1's stability. Indeed, when we replaced this glycine with a tyrosine by CRISPR/Cas9 gene editing (*htp-1$^{G97Y}$*), this mutant phenotypically resembled the *htp-1(me84)* null allele [11] in which this glycine residue is mutated to a glutamate, producing very few viable progeny and a high incidence of male (HIM) phenotype. Given that the severity of the *htp-1$^{G97Y}$* mutation would preclude our analysis of meiotic prophase events, we sought to create a less severe hypomorphic allele of *htp-1* to test whether it genetically interacts with the *pch-2* null mutation.

To identify a mutation that would allow analysis of pairing, synapsis, recombination, and meiotic progression, we used CRISPR/Cas9 to substitute glycine 97 with amino acids that progressively add bulkier side chains at this position (*htp-1$^{G97A}$*, *htp-1$^{G97S}$*, and *htp-1$^{G97T}$*). Upon initial examination of these strains, we observed a HIM phenotype associated with the *htp-1$^{G97T}$* mutant strain (Table 1), suggesting that this mutation causes meiotic chromosome

misegregation. This HIM phenotype was accompanied by a reduction in embryonic viability (Table 1). Therefore, we limited our analysis of testing genetic interactions with *pch-2* for pairing, synapsis, and recombination to *htp-1$^{G97T}$* mutants. Indirect immunofluorescence showed no difference in the loading of HTP-1 to meiotic chromosomes in *htp-1$^{G97T}$* mutants or in *htp-1$^{G97T}$ htp-2* double mutants (S4 Fig).

We first tested whether pairing is disrupted in this strain by performing immunofluorescence against HIM-8 in *syp-1*, *pch-2;syp-1*, *htp-1$^{G97T}$;syp-1* and *pch-2;htp-1$^{G97T}$;syp-1* mutant germlines (Fig 5A and 5B). In *htp-1$^{G97T}$;syp-1* double mutants, the percent of nuclei that have completed pairing in zone 2 is similar compared to *syp-1* single mutants (Fig 5B, p value = 0.553 by two-tailed Fisher's exact test), indicating that *htp-1$^{G97T}$;syp-1* double mutants do not have a defect in the progression of pairing. *pch-2;htp-1$^{G97T}$;syp-1* triple mutants have a similar rate of pairing in zone 2 compared to *pch-2;syp-1* double mutants (Fig 5B, p value = 0.311 by two-tailed Fisher's exact test), showing that the *htp-1$^{G97T}$* mutation does not suppress the acceleration of pairing observed in *pch-2* mutants.

## *htp-1$^{G97T}$* causes severe synapsis defects that are not suppressed by mutation of *pch-2*

Next, we tested whether synapsis is perturbed in *htp-1$^{G97T}$* mutants and whether this allele genetically interacts with the *pch-2* null allele (Fig 5C and 5D). Synapsis is severely disrupted in *htp-1$^{G97T}$* mutants: despite some chromosomes completing synapsis (Fig 5D), almost all meiotic nuclei have some number of unsynapsed chromosomes throughout the germline, where only 1.3% of nuclei reach complete synapsis (Fig 5D). In *pch-2;htp-1$^{G97T}$* double mutants, we find that synapsis is disrupted as severely as *htp-1$^{G97T}$* single mutants, where 1.3% of nuclei also reach complete synapsis (Fig 5D, p value = 0.717 by two-tailed Fisher's exact test), indicating that these alleles do not genetically interact and regulate synapsis independent of each other.

## *htp-1$^{G97T}$* leads to a delay in RAD-51 removal that is not suppressed by mutation of *pch-2*

Next, we assessed the progress of DNA repair in *htp-1$^{G97T}$* and *pch-2;htp-1$^{G97T}$* mutants by scoring RAD-51 foci throughout the germline (Fig 5E and 5F). In comparison to control germlines, *htp-1$^{G97T}$* single mutants exhibit a delay in the installation of RAD-51 foci (Fig 5F). However, as meiotic prophase continues, meiotic nuclei in *htp-1$^{G97T}$* single mutants have more RAD-51 foci and substantially delay their removal, possibly as an indirect consequence of the defects in synapsis we observe in this mutant (Fig 5D). *pch-2;htp-1$^{G97T}$* double mutants no longer exhibit this delay in RAD-51 installation, suggesting a partial suppression of this particular phenotype, but display more RAD-51 foci, particularly in zones 4 and 5, than wildtype meiotic nuclei and *pch-2* single mutants, similar to *htp-1$^{G97T}$* single mutants. Thus, PCH-2 and HTP-1 do not appear to genetically interact to regulate the rate of DNA repair.

## *htp-1$^{G97T}$* causes defects in crossover recombination that are not suppressed by mutation of *pch-2*

To continue our analysis, we analyzed crossover recombination by scoring the number of GFP::COSA-1 foci per nucleus in wildtype, *pch-2*, *htp-1$^{G97T}$*, and *pch-2;htp-1$^{G97T}$* mutants (Fig 5G and 5H). Surprisingly, we found that in *htp-1$^{G97T}$* single mutants, the number of COSA-1 foci varies greatly across nuclei, ranging from 4 to 9, with only 39% of nuclei having 6 GFP::COSA-1 foci (Fig 5H), suggesting that the *htp-1$^{G97T}$* mutation causes misregulation of

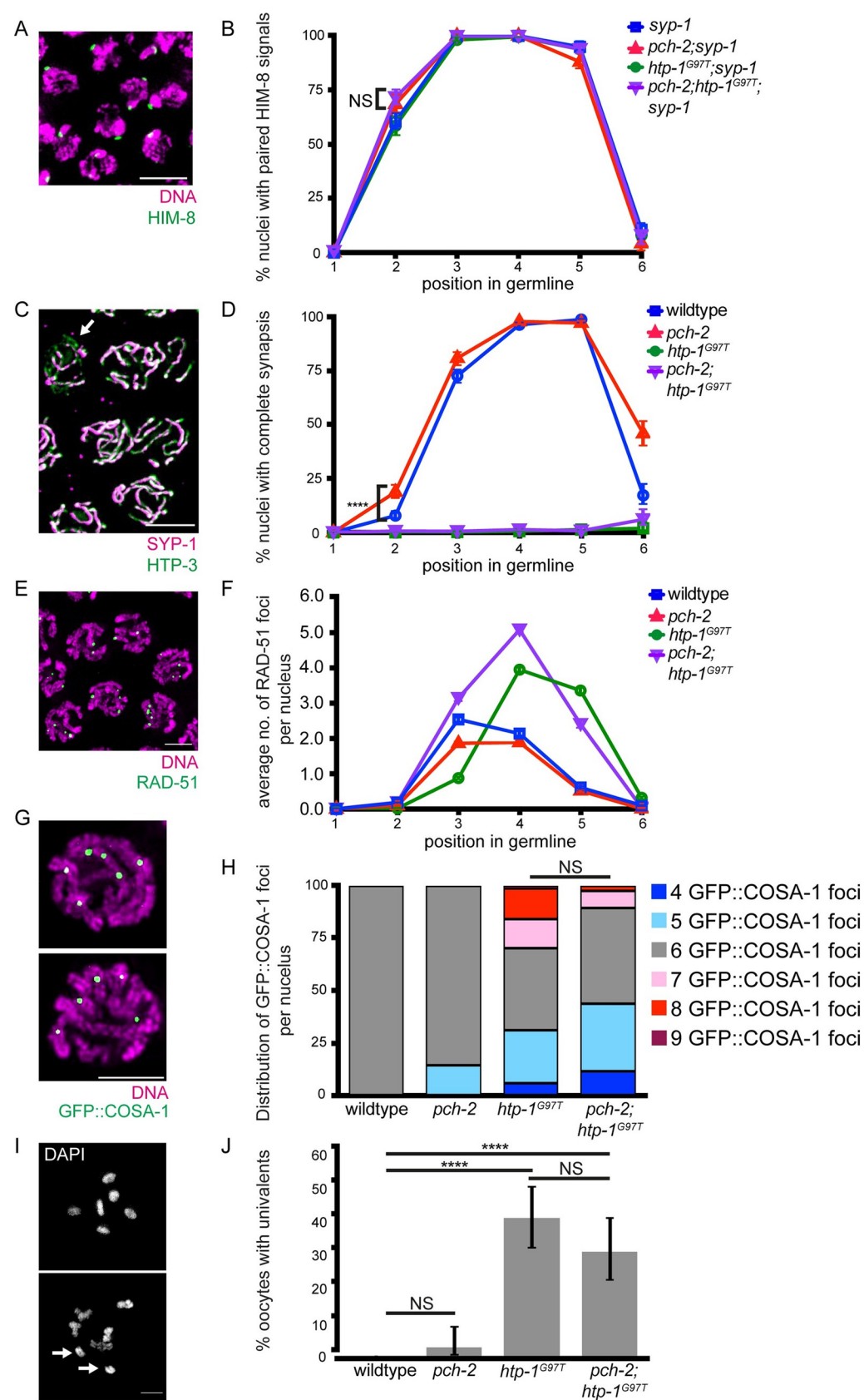

**Fig 5. *htp-1^{G97T}* does not genetically interact with *pch-2* in the context of pairing, synapsis, DNA repair, or crossover formation. A.** Meiotic nuclei in zone 2 in *syp-1* mutants stained with DAPI (magenta) and antibody against HIM-8 (green). **B.** Quantification of percent nuclei with paired HIM-8 signals in zones 1–6 for *syp-1* (blue), *pch-2;syp-1* (red), *htp-1^{G97T};syp-1* (green) and *pch-2;htp-1^{G97T};syp-1* (purple) mutant strains. Error bars represent 95% confidence intervals. **C.** Meiotic nuclei in zone 3 in wildtype worms stained with antibodies against SYP-1(magenta) and HTP-3 (green). Arrow indicates unsynapsed chromosomes. **D.** Quantification of percent nuclei with complete synapsis in zones 1–6 for wildtype (blue), *pch-2* (red), *htp-1^{G97T}* (green) and *pch-2;htp-1^{G97T}* (purple) mutant strains. Error bars represent 95% confidence intervals. **E.** Meiotic nuclei in zone 3 in wildtype worms stained with DAPI (magenta) and antibody against RAD-51 (green). **F.** Quantification of average number of RAD-51 foci in zones 1–6 for wildtype (blue), *pch-2* (red), *htp-1^{G97T}* (green) and *pch-2;htp-1^{G97T}* (purple) mutant strains. Error bars represent standard error of the mean (SEM). **G.** Meiotic nuclei in control and *pch-2* mutant worms stained with DNA (magenta) and antibodies against GFP:: COSA-1 (green). Top indicates representative nuclei with 6 GFP::COSA-1 foci. Bottom indicates representative nuclei with 5 GFP::COSA-1 foci. **H.** Quantification of distribution of GFP::COSA-1 foci per nucleus for wildtype (n = 298), *pch-2* (n = 334), *htp-1^{G97T}* (n = 241) and *pch-2;htp-1^{G97T}* (n = 254)) mutant strains. **E.** Oocytes from wildtype and *pch-2* mutant worms stained with DAPI. Top indicates representative oocyte with zero univalents. Bottom indicates representative oocyte with univalents (arrows). **F.** Quantification of percent oocytes with univalents for wildtype (n = 94), *pch-2* (n = 92), *htp-1^{G97Y}* (n = 104), and *pch-2;htp-1^{G97Y}* (n = 92) mutant strains. In all graphs, NS indicates non-significant and **** indicates p-value < 0.0001. Significance was assessed using two-tailed Fisher's exact tests. In all images, scalebar indicates 5 microns.

crossover formation. In addition to the loss of crossover assurance (nuclei with less than 6 GFP::COSA-1 foci), we also observe a loss of crossover interference (nuclei with greater than 6 GFP::COSA-1 foci), indicating a general loss of crossover control in *htp-1^{G97T}* mutants. In *pch-2;htp-1^{G97T}* double mutants, we observe a similar phenotype where the number of GFP:: COSA-1 varies across nuclei, ranging from 4 to 9, with only 45% having 6 COSA-1 foci (Fig 5H, p value = 0.1725 by two-tailed Fisher's exact test). Overall, these data indicate that PCH-2 and HTP-1 do not genetically interact to regulate crossover recombination.

We further analyzed crossover formation by looking directly at bivalents in oocytes and quantified the frequency of univalents in wildtype, *pch-2*, *htp-1^{G97T}*, and *pch-2;htp-1^{G97T}* mutants (Fig 5I and 5J). In *htp-1^{G97T}* single mutants, we found that the frequency of univalents significantly increases to 41% (Fig 5J, p value < 0.0001 by two-tailed Fisher's exact test), further verifying the defect in crossover formation in *htp-1^{G97T}* single mutants. For *pch-2;htp-1^{G97T}* double mutants, we also found that the frequency of univalents is statistically higher compared to wildtype oocytes (31%, Fig 5J, p value < 0.0001 by two-tailed Fisher's exact test) but not significantly different when compared to *htp-1^{G97T}* single mutants (31%, Fig 5J, p value = 0.182 by two-tailed Fisher's exact test). Taken together, this suggests that both *pch-2* and *htp-1* both promote crossover assurance but regulate it through different mechanisms.

## *htp-1^{G97T}* delays meiotic progression and this delay is partially suppressed by mutation of *pch-2*

The absence of HTP-1 accelerates meiotic progression, producing non-homologous synapsis and more rapid DNA repair [11]. In contrast, mutations within the HORMA domain of HTP-1 that prevent its loading on meiotic chromosomes cause a delay in meiotic progression, producing asynapsis [53]. We tested whether *htp-1^{G97T}* mutations similarly affect meiotic progression by performing immunostaining against DSB-1.

We quantified the fraction of DSB-1 positive meiotic nuclei in wildtype and *htp-1^{G97T}* mutant germlines (Fig 6A). We found that *htp-1^{G97T}* mutants display a statistically significant increase in the fraction of DSB-1 positive nuclei (81%, p value < 0.0001 by two-tailed Fisher's exact test Fig 6B), indicating that this mutation produces a delay in meiotic progression. Since DSB-1 is required for DNA double-strand break formation, the extension of DSB-1 likely explains the increase in RAD-51 foci (Fig 5F) and the attenuation of crossover interference (Fig 5H) in *htp-1^{G97T}* mutants. Moreover, since the increase in RAD-51 foci formation

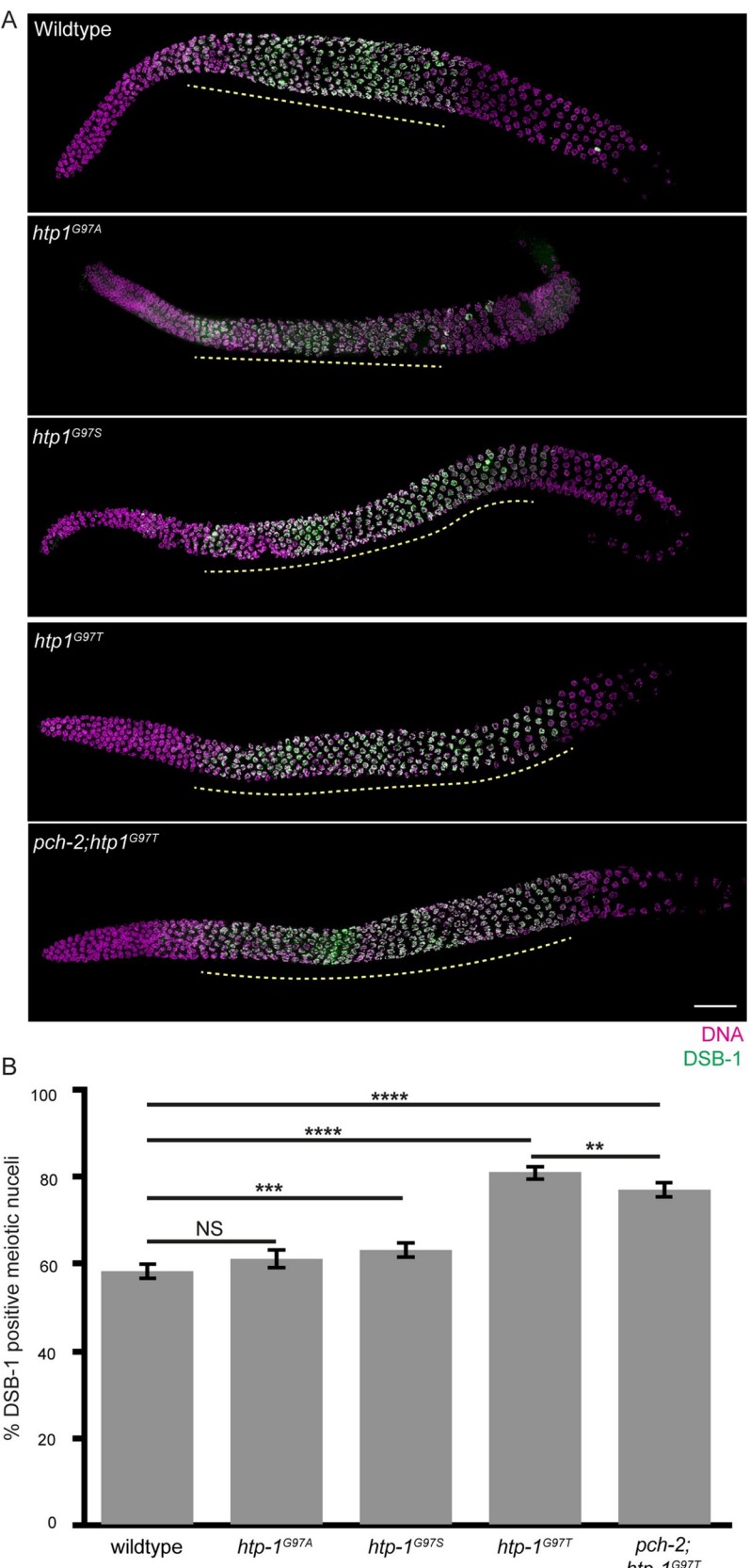

**Fig 6. $htp\text{-}1^{G97T}$ delays meiotic progression and this delay is partially suppressed by mutation of $pch\text{-}2$. A.** Full length representative germlines stained with DAPI (magenta) and antibody against DSB-1 (green). Yellow dashed line represents region of DSB-1 positive nuclei. Scalebar indicates 20 microns. **B.** Quantification of DSB-1 positive meiotic nuclei for wildtype(n = 2355), $htp\text{-}1^{G97A}$ (n = 2340), $htp\text{-}1^{G97S}$ (n = 2113), $htp\text{-}1^{G97T}$ (n = 1955) and $pch\text{-}2;htp\text{-}1^{G97T}$ (n = 1763) mutant strains. NS indicates non-significant. ** indicates p-value <0.01. *** indicates p-value < 0.001. **** indicates p-value <0.0001. Statistical significance was assessed using two-tailed Fisher's exact tests. Error bars represent 95% confidence intervals.

(Fig 5F) is accompanied by an increase in GFP::COSA-1 foci (Fig 5H), $htp\text{-}1^{G97T}$ mutants also appear to have defects in crossover homeostasis, a mechanism that ensures that the number of crossovers remain invariant even when the number of crossover precursors, such as DNA double-strand breaks, vary [54] and which is particularly robust in *C. elegans* [47].

Given the severity of this phenotype, we wondered if the less severe mutations we generated, $htp\text{-}1^{G97A}$ and $htp\text{-}1^{G97S}$ mutants, might also affect meiotic progression, albeit more weakly (Fig 6A). While $htp\text{-}1^{G97A}$ mutants did not affect the fraction of DSB-1 positive nuclei, $htp\text{-}1^{G97S}$ mutants produced a small but statistically significant delay in meiotic progression (63% DSB-1 positive nuclei, Fig 6B, p value = 0.0006 by two-tailed Fisher's exact test), indicating that the introduction of bulkier amino acids at this residue produces an allelic series with respect to defects in meiotic progression.

We next tested whether this defect in meiotic progression in $htp\text{-}1^{G97T}$ single mutants is suppressed by the $pch\text{-}2$ mutation. We found that the percent of DSB-1 stained nuclei is slightly but significantly lower in $pch\text{-}2;htp\text{-}1^{G97T}$ double mutants (77%, Fig 6B) compared to $htp\text{-}1^{G97T}$ single mutants (Fig 6B, p value = 0.0047 by two-tailed Fisher's exact test), indicating that the meiotic progression defect in $htp\text{-}1^{G97T}$ mutants is partially suppressed by mutations in $pch\text{-}2$ and that these two proteins cooperate to regulate meiotic progression (Table 2).

## $him\text{-}3^{R93Y}$ and $htp\text{-}1^{G97T}$ mutants have functional meiotic checkpoints

We previously showed that $htp\text{-}3^{H96Y}$ mutants abolishes both the DNA damage response and the synapsis checkpoint [43], similar to null mutations in $htp\text{-}3$ and $him\text{-}3$ [55]. A null mutation in $htp\text{-}1$ only affects the synapsis checkpoint [55]. To determine whether $him\text{-}3^{R93Y}$ or $htp\text{-}1^{G97T}$ mutants affect meiotic checkpoint activation, we introduced each mutation into the $syp\text{-}1$ mutant background and evaluated the levels of germline apoptosis. $syp\text{-}1$ mutants activate both the DNA damage response and the synapsis checkpoint, producing high levels of germline apoptosis [48]. $him\text{-}3^{R93Y}$ mutants showed wildtype levels of apoptosis while $htp\text{-}1^{G97T}$ mutants exhibited germline apoptosis similar to that of $syp\text{-}1$ single mutants (S6 Fig, p value < 0.0001 by student t-test), likely as a consequence of the asynapsis and defects in recombination we observe (Fig 5). Both $syp\text{-}1;him\text{-}3^{R93Y}$ and $syp\text{-}1;htp\text{-}1^{G97T}$ double mutants showed similar levels of germline apoptosis as $syp\text{-}1$ single mutants (S6 Fig), indicating that both $him\text{-}3^{R93Y}$ or $htp\text{-}1^{G97T}$ mutants are competent for meiotic checkpoint activation, unlike $htp\text{-}3^{H96Y}$ mutants [43].

## $him\text{-}3^{R93Y}$ mutants maintain the structure of the HORMA domain but have reduced affinity for closure motifs

Our data thus far demonstrates that $pch\text{-}2$ genetically interacts with specific alleles of $htp\text{-}3$, $him\text{-}3$ and $htp\text{-}1$, with varying effects on pairing, synapsis, recombination and meiotic progression. However, how these genetic interactions translate into a molecular understanding of the functional relationship between PCH-2 and these meiotic HORMADs is unclear. To address this, we tested what effect these mutations had on the ability of meiotic HORMADs to bind their closure motifs and adopt the closed conformation.

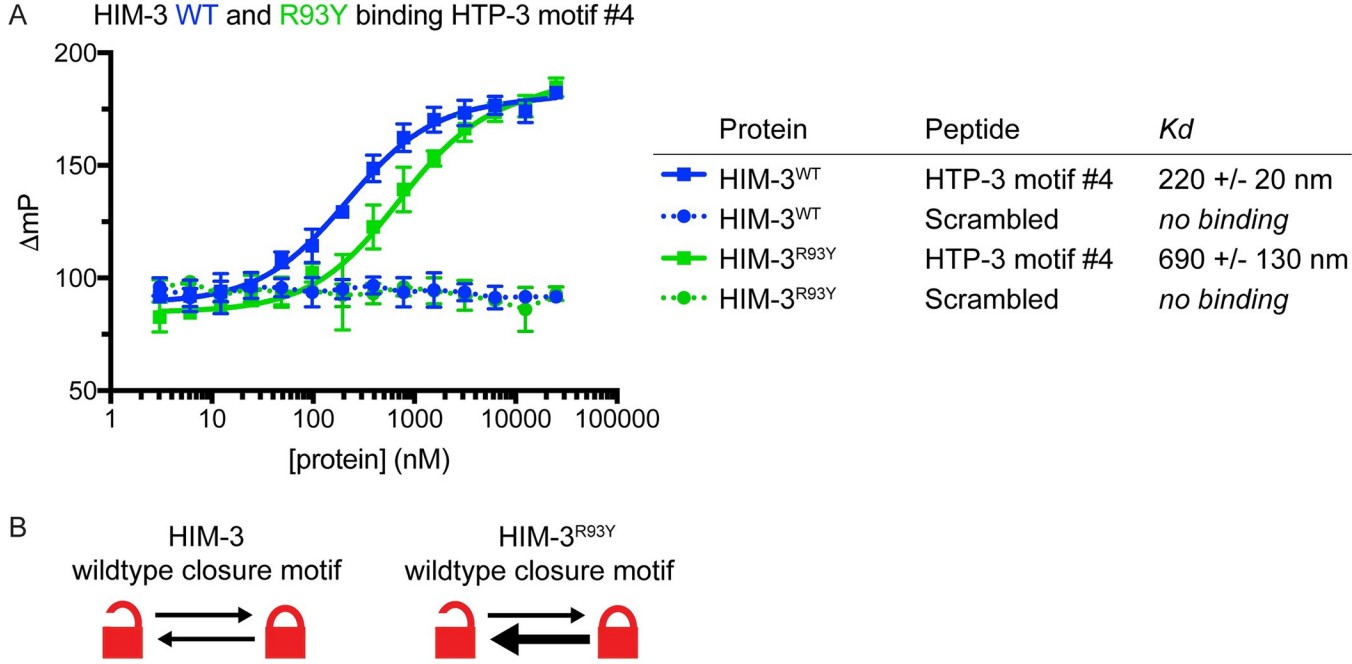

**Fig 7. him-3[R93Y] mutants have reduced affinity for closure motifs. A.** Fluorescence polarization peptide binding assay of HIM-3 (blue) and HIM-3[R93Y] (green) and measured Kd's. **B.** Cartoon depicting proposed model for open and closed conformation switching of wildtype HIM-3 and HIM-3[R93Y].

While the closure motif for HTP-3 is unknown, HIM-3 is known to bind closure motifs found within the C-terminal tail of HTP-3 [19]. HIM-3 also contains a closure motif in its own C-terminus but it preferentially binds closure motifs in HTP-3 to contribute to a hierarchical HORMAD assembly [19]. We cloned and expressed wildtype HIM-3 and HIM-3[R93Y] in *E. coli*, purified each protein, and performed a fluorescence polarization peptide binding assay to test how well HIM-3[R93Y] binds a peptide encoding HTP-3 closure motif #4 *in vitro*. Consistent with prior work, we found that wildtype HIM-3 binds the HTP-3 closure motif with a $K_d$ of 220 +/- 20 nM and found that it does not bind to a control peptide with a scrambled sequence (Fig 7A). HIM-3[R93Y] binds to the same closure motif with an increased $K_d$ of 690 +/- 130 nM, indicating that HIM-3[R93Y] binds the HTP-3 closure motif with reduced affinity compared to wildtype HIM-3 (Fig 7A). The thermal stability of HIM-3[R93Y] is similar to that of wild-type HIM-3 (S5 Fig), and a crystal structure of HIM-3[R93Y] in the closed conformation is equivalent to that of wildtype HIM-3 (Fig 2D), suggesting that reduced affinity for the closure motif for HIM-3[R93Y] is not due to instability of the protein or an inability to adopt the closed conformation. These results demonstrate that HIM-3[R93Y] binds the HTP-3 closure motif less readily than wildtype HIM-3, which would potentially limit its ability to adopt the closed conformation *in vivo* (Fig 7B). Because this residue is positioned similarly within all three-essential meiotic HORMADs (Fig 2B), our biochemical analysis of HIM-3[R93Y] may be generalizable to the mutant versions of HTP-3 and HTP-1: HTP-3[H96Y] and HTP-1[G97T] could bind their respective closure motifs with less affinity, affecting their ability to adopt the closed conformation *in vivo*.

Unlike the budding yeast meiotic HORMAD, Hop1 [56], HIM-3 is difficult to purify in any conformation aside from its closed conformation [19], preventing us from directly assessing whether the HIM-3[R93Y] mutation affects the protein's closed-versus-unbuckled conformational equilibrium. However, the reduced affinity of HIM-3[R93Y] for the HTP-3 closure motif, when combined with the established conformational dynamics of Hop1 [56], the localization

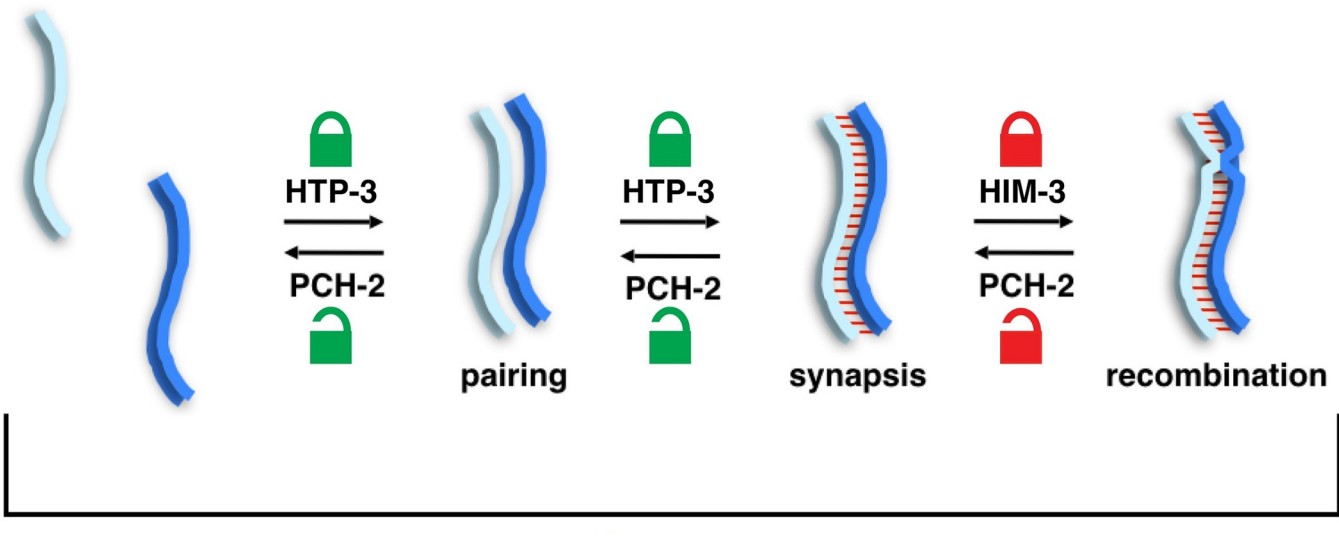

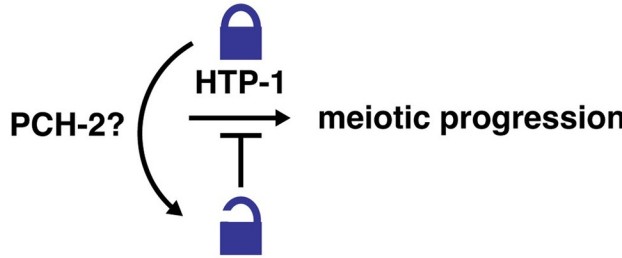

**Fig 8. PCH-2 distributes its regulation of meiotic prophase events by remodeling different meiotic HORMADs.** Cartoon depicting proposed model for PCH-2's regulation of meiotic prophase events. PCH-2 regulates the progression of pairing and by acting through HTP-3, crossover recombination through HIM-3, and may regulate meiotic progression through HTP-1.

of PCH-2 by structured illumination microscopy (Fig 1) and our genetic and cytological analysis (Figs 3–6) of how these mutant alleles suppress some defects observed in *pch-2* mutants, presents a model consistent with PCH-2 remodeling meiotic HORMADs from a closed to unbuckled conformation on meiotic chromosomes to control the progress of pairing, synapsis, meiotic recombination and cell cycle progression (Fig 8).

## Discussion

Since the discovery of Pch2 in budding yeast more than 20 years ago [38], the particular roles of Pch2/PCH-2/TRIP13 in meiosis have been enigmatic. More recently, biochemical and structural analysis of how mammalian TRIP13 contributes to spindle checkpoint progression by remodeling the HORMA domain protein MAD2 [24,26,27] have provided a useful framework to establish Pch2/PCH-2/TRIP13's diverse roles in structuring meiotic chromosomes for homolog pairing, synapsis, and recombination, and signaling meiotic progression. In the spindle checkpoint, PCH-2/TRIP13 promotes checkpoint activation by maintaining a pool of soluble "open" MAD2, which is used to build the mitotic checkpoint complex at unattached kinetochores and delay anaphase onset [57–59]. After kinetochore attachment, TRIP13 plays an additional role in human cells, disassembling the mitotic checkpoint complex by remodeling MAD2, promoting checkpoint inactivation and anaphase progression [60,61]. This activity may not necessarily be conserved, given that a similar role in silencing the checkpoint is not observed in *C. elegans* [59,62], and Pch2 is not expressed outside meiosis in *S. cerevisiae* [38].

Nonetheless, in some systems, TRIP13 appears to play dual roles in spindle checkpoint activation and inactivation, through a single biochemical activity of converting "closed" MAD2 to the "open" conformer.

Recently, studies of Pch2 in budding yeast, and by extension plants and mammals, have suggested that this enzyme plays a similar dual role in meiotic chromosome organization, homologous recombination, and meiotic progression. Meiotic HORMADs in most organisms adopt a "self-closed" conformation with their HORMA domains bound to a closure motif in their own C-terminus [19,28,30]. In budding yeast and plants, Pch2/TRIP13's remodeling activity is needed to convert these "self-closed" HORMADs into an "open" or "unbuckled" conformation to promote their nuclear import and association with chromosomes [29,30]. After assembly into axial elements of meiotic chromosomes, HORMADs promote meiotic recombination and homolog pairing [6,63], likely through direct protein-protein interactions with partner proteins [23]. Once homologs have recombined and synapsed, Pch2/TRIP13 is recruited to the synaptonemal complex [34,35,38], where it is thought to disassemble HORMAD-axis protein complexes to deplete them [10,33,35]. HORMAD depletion down-regulates recombination, promotes rapid repair of remaining DNA breaks, and facilitates meiotic progression [64,65].

In *C. elegans*, several key differences in the timing of meiotic events relative to yeast, plants, and mammals necessitate a rethinking of PCH-2's roles in meiotic progression. First, homolog pairing and synapsis occur independently of recombination in *C. elegans* [66], such that synapsis cannot serve as a signal for successful interhomolog recombination as in other organisms. Second, meiotic HORMADs localize to meiotic chromosomes independently of PCH-2 [2] and are not visibly depleted in response to synapsis [21,22,36], indicating the dual role for Pch2/TRIP13 proposed in other systems is not relevant to understanding PCH-2's function in *C. elegans* meiosis. Finally, a null mutation of *pch-2* or mutations that eliminate its ATP hydrolysis activity result in accelerated homolog pairing, synapsis, and meiotic recombination, leading to a loss of meiotic fidelity [2,3]. Because these events are likely mediated by direct HORMAD-partner protein interactions on meiotic chromosomes, and because PCH-2 is localized to chromosomes during the relevant stages of meiotic prophase (Fig 1) [2], these phenotypes suggest that PCH-2 either reduces overall levels of HORMADs on chromosomes or increases their association-dissociation dynamics to modulate these key events.

Our experiments from this study shed further light on this model. We have identified mutations in the HORMA domains of both HTP-3 and HIM-3 (Fig 2A) that appear to load normally on chromosomes [42] (S2 and S4 Figs), reduce the affinity of closure motif interactions in vitro (Fig 7) and suppress specific defects of a *pch-2* null mutation in homolog pairing/synapsis and recombination, respectively (Figs 3 and 4). These data lead us to propose a model in which PCH-2 remodels the HORMA domains of chromosome-associated HORMADs from the "closed" conformation to the "open/unbuckled" conformation to temporarily reduce HORMAD occupancy on chromosomes, decelerating HORMAD-dependent homolog pairing, synapsis, and recombination to promote meiotic fidelity (Fig 8). An attractive but still speculative extension to this model is that PCH-2's remodeling of meiotic HORMADs destabilizes HORMAD-partner protein interactions essential for homolog pairing, synapsis, and recombination, deliberately slowing their progress.

Our model has important implications for the established roles of Pch2/TRIP13 in other organisms. First, Pch2/PCH2 in both yeast and plants localize specifically to sites associated with synapsis initiation and recombination [34,35], suggesting a similar role temporarily reducing HORMAD occupancy at these sites prior to synapsis. Further, despite the observation that Pch2/TRIP13 strongly depletes meiotic HORMADs from chromosomes upon synapsis, these proteins are not completely removed from chromosomes in any organism [10,33,35],

suggesting roles for both meiotic HORMADs and Pch2/TRIP13 even after chromosome synapsis. Given that the gradual implementation of crossover formation partially overlaps when chromosomes are synapsed in budding yeast, plants, and mice [67–70], we propose that the presence of Pch2/TRIP13 on synapsed chromosomes remodels meiotic HORMADS to regulate the number and location of crossovers, similar to one role of PCH-2 in *C. elegans* [2,71].

The relationship between PCH-2 and HTP-1 is less clear. Despite the small structural differences between HIM-3 and HTP-1 (Fig 2E) that required some adjustments in how we designed our mutational analysis of HTP-1, we predict that HTP-1$^{G97T}$ also shows a reduced affinity for its closure motifs in both HTP-3 and HIM-3, potentially shifting its equilibrium toward the unbuckled conformation. The consequences of this difference in protein function are a significant delay in meiotic progression (Fig 6B) and defects in synapsis, DNA repair and crossover recombination (Fig 5D, 5F, and 5H), consistent with previous data [11,53,72]. However, given our data, we favor the interpretation that the primary defect in *htp-1$^{G97T}$* mutants is the delay in meiotic progression, with secondary, indirect consequences for synapsis and recombination. Mutation of *pch-2* partially suppresses this defect in meiotic progression (Fig 6B), with little effect on synapsis or recombination (Fig 5D and 5H), supporting our interpretation. How might a delay in meiotic progression prevent complete synapsis or the normal completion of recombination? This meiotic delay might maintain a cell cycle state in which the synaptonemal complex and/or recombination intermediates are more labile and subject to turnover, through mechanisms such as post-translational modifications, high chromosome mobility, some other unknown condition or some combination of these conditions. Finally, if we are correct and an increase in unbuckled HTP-1$^{G97T}$ is responsible for a meiotic delay, with indirect consequences on synapsis and recombination, our data also suggest that, unlike open MAD2, unbuckled versions of meiotic HORMADs may have additional functions aside from being an important structural intermediate poised for binding to closure motifs in partner proteins.

That mutation of *pch-2* only weakly suppresses the delay in meiotic progression we observe in *htp-1$^{G97T}$* mutants raises two possibilities: 1) the *htp-1$^{G97T}$* mutation is too severe to allow us to observe this genetic interaction; or 2) PCH-2 does not directly act on HTP-1 to regulate meiotic progression. How might PCH-2 contribute to the regulation of meiotic progression without directly remodeling HTP-1? Since both HTP-3 and HIM-3 have HTP-1 closure motifs in their C-termini [19], PCH-2's remodeling of HTP-3 and HIM-3 may indirectly result in the removal, and remodeling of HTP-1, contributing to the regulation of meiotic progression. This possibility is particularly attractive since it explicitly coordinates meiotic progression with the rate of pairing, synapsis and recombination.

Finally, our work shows that PCH-2's regulation of pairing, synapsis, recombination and meiotic progression are cleanly delegated to the remodeling of different meiotic HORMADs: We propose that PCH-2 remodels HTP-3 from a closed to an unbluckled conformation to decelerate pairing and synapsis and contribute to checkpoint activation, HIM-3 to decelerate crossover recombination and HTP-1, either directly or indirectly, to control meiotic progression. This deceleration allows for proofreading of the homolog interactions that underlie these events and ensure fidelity [3]. Similar separation of function has been observed in plants and mice: each have two meiotic HORMADs, only one of which (HORMAD1 in mice, ASY1 in plants), appears to be essential for pairing, synapsis and recombination [9,15–17,73]. In the case of mice, a second HORMAD, HORMAD2, is not required for pairing, synapsis or recombination but is essential for efficient meiotic checkpoint responses [18,74]. In plants, no role for ASY2 has been reported yet. The separation of function we observe in *C. elegans* is slightly different: The three essential meiotic HORMADs have overlapping but distinct roles that nonetheless significantly impact pairing, synapsis and recombination [11–13,19,21,22].

However, it is through their regulation by PCH-2 that their distinct roles become more clear, allowing for the coordination of pairing, synapsis, recombination and meiotic progression. This delegation of tasks, as revealed by their regulation by PCH-2, may also explain why this family has expanded so dramatically in *C. elegans*. Since multiple systems, including mammals, also have more than one meiotic HORMAD, this delegation of tasks may be a conserved evolutionary feature of meiosis.

Another possible contributing factor to the expansion of this family in *C. elegans* is the mechanistic uncoupling of meiotic recombination with synapsis. Mice, budding yeast and plants all rely on the initiation of meiotic recombination [75–78] to promote accurate synapsis. Similarly, all of these organisms have a single essential meiotic HORMAD controlling pairing, synapsis and recombination [6,9,15], suggesting a possible connection between these observations. In contrast to these systems, *C. elegans* does not rely on meiotic recombination for accurate chromosome synapsis [66], suggesting that the family of meiotic HORMADs may have expanded to accommodate the fact that synapsis and recombination are regulated by two different mechanisms in this system.

While *C. elegans* is unique in that it has four meiotic HORMADs which assemble into a hierarchical complex, it is formally possible that other systems also have different functional classes of meiotic HORMADs, even when they have a single meiotic HORMAD in their genome. For example, in budding yeast, Hop1 can bind directly to nucleosomal DNA [79], a closure motif in the axis protein Red1 and closure motifs in Hop1 once it has bound to Red1, producing oligomerization [28]. These three different classes of Hop1, potentially defined by the molecular nature of their binding to meiotic chromosomes, may also assemble hierarchically and have different effects on the events of meiotic prophase. These differences could affect their regulation by Pch2, producing a similar delegation of functions as we observe with HTP-3, HIM-3 and HTP-1. For example, it has already been shown in budding yeast that in regions adjacent to telomeres, Hop1 is resistant to Pch2-mediated redistribution [80], consistent with this possibility of differential regulation by Pch2.

Our data suggest that the remodeling of meiotic HORMADs by PCH-2 has consequences on the progress, coordination, and fidelity of meiotic prophase events. Major questions that arise from this work are 1) how is PCH-2 activity limited to meiotic HORMADs in some contexts but not others and 2) how do these meiotic HORMADs contribute to the events of pairing, synapsis and crossover recombination? How does HTP-3 molecularly participate in pairing and synapsis? How does HIM-3 promote crossover recombination? And how does HTP-1 control meiotic progression? Future studies focused on understanding how meiotic HORMADs molecularly interact with each other, other axis proteins, and/or additional binding partners will shed new light on how meiotic HORMADs, in collaboration with PCH-2, drive the coordination of meiotic prophase events to promote fidelity and prevent aneuploidy.

## Materials and methods

### Genetics and worm strains

The C. *elegans* Bristol N2 strain was used as the wildtype strain throughout this study [81]. All strains were grown at 20˚C under standard conditions unless otherwise stated. Mutations and rearrangements used were as follows:

LG I: *htp-3(vc75), mnDp66*

LG II: *pch-2(tm1458), meIs8 [Ppie-1::GFP::cosa-1 + unc-119(+)]*

LG IV: *him-3(blt9), htp-1(blt10), htp-1(blt11), htp-1(blt12), htp-2(tm2543), nT1[unc-?(n754let-?(m435)]*

LG V: *syp-1(me17), bcIs39[Plim-7::ced-1::gfp; lin-15(+)]*

LG X: *meDf2*

*meDf2* is a terminal deficiency of the left end of the X chromosome that removes the X chromosome pairing center (PC) as well as numerous essential genes. For this reason, homo- and hemizygous *meDf2* animals also carry a duplication (*mnDp66*) that includes these essential genes but does not interfere with normal X chromosome segregation [82].

The *him-3^R93Y^* (*him-3[blt9]*), *htp-1^G97A^* (*htp-1[blt10]*), *htp-1^G97S^* (*htp-1[blt11]*), and *htp-1^G97T^* (*htp-1[blt12]*) were generated by CRISPR-Cas9 genome editing supplemented with the *dpy-10* co-conversion method [83].

The following guide RNA sequences were used:

*dpy-10* sgRNA: 5'GCU ACC AUA GGC ACC ACG AGG UUU UAG AGC UAU GCU 3'
*him-3* sgRNA: 5'CGU GUG UCU UCA ACA UUC GAG UUU UAG AGC UAU GCU 3'
*htp-1* sgRNA: 5' UCA ACU ACU UCG AAA UGC UGG UUU UAG AGC UAU GCU 3'

The following DNA repair oligonucleotide sequences were used:

*dpy-10*: 5' CAC TTG AAC TTC AAT ACG GCA AGA TGA GAA TGA CTG GAA ACC GTA CCG CAT GCG GTG CCT ATG GTA GCG GAG CTT CAC ATG GCT TCA GAC CAA CAG CCT AT 3'

*him-3^R93Y^*: 5' GCA TTC AGA GGC AGT GTT CCT GCC GCA AAG CGT GTG TCT TCA ACA TTT GAC GGA TTG TAC GAT GCG ATT CAA CAA GGC TAT TTG CGA GAG TTC GCA ATC GTG TTC TAC AAG 3'

*htp-1^G97A^*: 5' GCT TCA GAA ACC CAC GGT CTA ACC AAA TCG CTC AAC TAC TTC GAA ATG CTG CTG ATG CAA CAA AGG ATG GGT TTC TGA AAG AAG TCT CCC TCG TGA TCA CAA ATA ATG 3'

*htp-1^G97S^*: 5' GCT TCA GAA ACC CAC GGT CTA ACC AAA TCG CTC AAC TAC TTC GAA ATG CTT CAG ATG CAA CAA AGG ATG GGT TTC TGA AAG AAG TCT CCC TCG TGA TCA CAA ATA ATG 3'

*htp-1 ^G97T^*: 5' 5' GCT TCA GAA ACC CAC GGT CTA ACC AAA TCG CTC AAC TAC TTC GAA ATG CTA CAG ATG CAA CAA AGG ATG GGT TTC TGA AAG AAG TCT CCC TCG TGA TCA CAA ATA ATG 3'

For CRISPR/Cas9 gene editing, all mutations were made in the N2 wildtype strain, with the exception of the *htp-1^G97T^ htp-2* double mutant strain, in which the *htp-1^G97T^* mutation was introduced in the *htp-2(tm2543)* mutant background. sgRNAs and oligo repair templates were synthesized by Integrated DNA technologies (IDT, Coraville, IA). Injection mixes contained either *htp-1* or *him-3* sgRNA (100 μM final), *dpy-10* sgRNA (100 μM final, [83]), tracrRNA (100 μM final, IDT), purified Cas9 protein (40 μM final), *htp-1* or *him-3* DNA repair oligo (50 μM final, IDT) and *dpy-10* repair oligo (50 μM final, IDT, [83]). Young adults were injected, recovered and stored at either 15 or 20 degrees. Roller or *dpy* F1's were singled to individual plates seeded with OP50 and screened for presence of the mutant allele by PCR and BspHI restriction digest (*htp-1)* or RsaI restriction digest (*him-3*). Individual F2's were then isolated from F1 plates that contained the allele to identify a homozygous strain.

All mutations were verified by sequencing and strains were backcrossed against wildtype worms three times.

For viability assays and progeny counts, the frequencies of hermaphrodites and males produced by a given genotype were determined by scoring the complete broods of twelve hermaphrodites worms transferred to individual plates at the L4 stage. Parents were transferred daily to facilitate accurate scoring of eggs and L1 larvae and progeny were scored as L4s or adults. When the parental genotype included *meDf2* heterozygotes, the numbers presented in Table 1 were adjusted to compensate for the inviability of *meDf2* homozygotes and hemizygotes that lack a duplication.

## Immunofluorescence, antibodies and microscopy

Day 1 adult hermaphrodites were dissected 24–28 hours post-late L4 stage and immunostaining and DAPI staining were performed similarly to [48]. Gonad dissections were performed in 1X EBT (250 mM Hepes-Cl, pH 7.4, 1.18 M NaCl, 480 mM KCl, 20 mM EDTA, and 5 mM EGTA) supplemented with 0.1% Tween 20 and 20 mM sodium azide. An equal volume of 2% formaldehyde in EBT (final concentration was 1% formaldehyde) was added and allowed to incubate under a coverslip for 5 minutes. The sample was mounted on HistoBond slides (75 X 25 X 1 mm), freeze-cracked, incubated in methanol at −20˚C for 1 minute and transferred to PBST (PBS with 0.1% Tween 20). After several washes of PBST, the samples were blocked for 30 minutes in 1% bovine serum albumin diluted in PBST. A hand-cut paraffin square was used to cover the tissue with 50 μl of antibody solution, diluted in block solution. Incubation was conducted in a humid chamber overnight at 4˚C. Slides were rinsed in PBST and then incubated for 2 hours at room temperature with fluorophore-conjugated secondary antibody at a dilution of 1:500 in PBST. Samples were rinsed several times and DAPI stained in PBST for 10 minutes. Samples were then mounted with mounting media (20 M N-propyl gallate [Sigma-Aldrich] and 0.14 M Tris in glycerol) with a no. 1 1/2 (22 mm$^2$) coverslip and sealed with nail polish.

For analyzing bivalents, the above protocol was implemented with the exception that hermaphrodites were dissected and DAPI stained 48 hours post late L4 stage.

The following primary antibodies were used for this study: rat anti-HIM-8 1:500 [44], guinea pig anti-HTP-3 1:250 [36], chicken anti-HTP-3 1:250 [36], rabbit anti-SYP-1 1:500 [40], guinea pig anti-SYP-1 1:500 [40], rabbit anti-HIM-3 1:500 [14], rabbit anti-HTP-1 1:400 [22], rabbit anti-RAD-51 1:250 (Novus Biologicals, Littleton, CO), guinea pig anti-DSB-1 1:250 [49], mouse anti-GFP 1:100 (Invitrogen, Waltham, MA) and rabbit anti-PCH-2 1:500 [2].

The following secondaries were used for this study: Alexa 488 anti-guinea pig (Invitrogen), Alexa 488 anti-rabbit (Invitrogen), Alexa 488 anti-mouse (Invitrogen), Cy3 anti-rabbit (Jackson Immunochemicals, West Grove, PA), Cy3 anti-rat (Jackson Immunochemicals), Cy3 anti-guinea pig (Jackson Immunochemicals), and Cy5 anti-chicken (Jackson Immunochemicals). All secondary antibodies were used at a 1:500 dilution.

To visualize meiotic nuclei and bivalents, DAPI was used at a dilution of 1:10,000.

All images except for those in Fig 1 were acquired using a DeltaVision Personal DV system (Applied Precision) equipped with a 100X N.A. 1.40 oil-immersion objective (Olympus), resulting in an effective XY pixel spacing of 0.064 or 0.040 μm. Three-dimensional image stacks were collected at 0.2-μm Z-spacing and processed by constrained, iterative deconvolution. For structured Illumination microscopy, images were obtained as 125 nm spaced Z stacks, using a 100x NA 1.40 objective on a DeltaVison OMX Blaze microscopy system, 3D-reconstructed and corrected for registration using SoftWoRx. Image scaling and analysis were performed using functions in the softWoRx software package. Projections were calculated by a maximum intensity algorithm. Composite images were assembled and some false coloring was performed with ImageJ.

Whole germline images were assembled in ImageJ as in [84] using pairwise stitching and/ or grid collection stitching [85]. Quantification of pairing, synapsis, and RAD-51 foci was performed as in [3] with a minimum of three germlines per genotype. The number of nuclei scored for each experiment is provided in S2 Table or in figure legends.

For analysis of the rate of pairing, "paired" nuclei were scored as a single focus, whether it was elongated or circular. "Unpaired" nuclei were scored as two completely separate foci that each had an individual border.

For analysis of the rate of synapsis, nuclei were considered "completely synapsed" when complete colocalization of HTP-3 and SYP-1 was observed. "Unsynapsed" nuclei were assigned when nuclei had stretches of HTP-3 with missing SYP-1 signal.

For the analysis of DSB-1, nuclei were scored as positive when DSB-1 signal was detected across the entire nucleus while nuclei scored as negative had completely absent DSB-1 staining. The percent of DSB-1 positive meiotic nuclei was aggregated across all three germlines per genotype. To perform the Fisher's Exact test, the total number of positive DSB-1 positive nuclei for each meiotic mutant was compared to the number of DSB-1 positive wildtype nuclei.

Scoring of germline apoptosis was performed as previously described in [48] in strains containing *bcIs39 [lim-7p::ced-1::GFP + lin-15(+)]* with the following exceptions: L4 hermaphrodites were allowed to age for 22–24 hours. They were then mounted under coverslips on 2% agarose pads containing 0.2mM levamisole and scored. A minimum of twenty-five germlines was analyzed for each genotype and the experiment was performed three times to ensure reproducibility. One representative experiment is provided.

Line scans measuring pixel intensity for Fig 1D were performed in ImageJ. Channels were split and the pixel intensities for each fluorophore was measured across the distance of the region of interest (Fig 1C, yellow line). Distance in pixels and normalized intensity were plotted using R (R core team) with the ggplot2 package [86].

## Protein expression and purification

The full-length HIM-3 gene (residues 2–291) was PCR-amplified from a *C. elegans* cDNA library and cloned into UC Berkeley Macrolab vector 2-CT (Addgene #29706), which encodes a TEV protease-cleavable His$_6$-MBP (maltose binding protein) tag. PCR-based mutagenesis was used to generate the R93Y mutant construct. Proteins were expressed in *E. coli* strain Rosetta2 pLysS (EMD Millipore) at 20°C for 16 hours, then harvested by centrifugation and resuspended in Buffer A (20 mM Tris-HCl pH 7.5, 10% glycerol, 2 mM β-mercaptoethanol) plus 300 mM NaCl and 10 mM imizadole. Clarified lysate was passed onto a Ni$^{2+}$ affinity column (5 mL HisTrap HP, Cytiva) in Buffer A plus 300 mM NaCl and 10 mM imizadole, the column was washed in Buffer A plus 300 mM NaCl and 20 mM imizadole, then protein was eluted in Buffer A plus 300 mM NaCl and 250 mM imizadole. Proteins were buffer-exchanged into Buffer A plus 300 mM NaCl and 20 mM imizadole using a centrifugal concentrator (Amicon Ultra, EMD Millipore), then incubated with 1:10 w/w ratio of purifed TEV protease [87] for 48 hours. The cleaved mixture was passed through Ni$^{2+}$ affinity resin a second time to remove uncleaved protein, free His$_6$-MBP tags, and His$_6$-tagged TEV protease, and the flow-through was concentrated and further purified by size exclusion chromatography (Superdex 200; Cytiva) in Buffer A plus 300 mM NaCl. Fractions were concentrated by centrifugal concentrator, aliquoted, and stored at -80°C.

## Fluorescence polarization binding assays

N-terminal FITC-Ahx labeled peptides were synthesized (BioMatik), resuspended in DMSO, then diluted into binding buffer (20 mM Tris pH 7.5, 300 mM NaCl, 10% glycerol, 1 mM DTT, 0.1% NP-40). Fifty μL reactions containing 50 nM peptide plus indicated amounts of HIM-3 (WT or R93Y) protein were incubated 60 minutes at room temperature, then fluorescence polarization was read in 384-well plates using a TECAN Infinite M1000 PRO fluorescence plate reader. All binding curves were done in triplicate. Binding data were analyzed with Graphpad Prism v. 8 using a single-site binding model.

## ThermoFluor assays

For protein stability measurement by ThermoFluor assay, purified proteins were diluted to 0.1 mg/mL in a buffer containing 20 mM Tris-HCl pH 7.5, 300 mM NaCl, 10% glycerol, and 1 mM DTT. 0.5 μL of SPYRO Orange dye (5,000X concentrate; Thermo Fisher Scientific #S6650) was mixed with 49.5 μL protein solution, and the mixture was sealed in an optically clear PCR plate. In a Bio-Rad CFX Connect real-time PCR system, fluorescence was measured in scan mode "FRET" during a 25˚-99˚ temperature ramp (0.5˚ intervals, 30 second hold per interval). For analysis, the change in fluorescence per step was plotted against temperature, and the peak of this plot (the representing the inflection point in the upward slope of the raw fluorescence curve) was taken as the melting temperature (Tm).

## Protein crystallization and structure determination

For crystallization, HIM-3(R93Y) was concentrated to 15–20 mg/mL in crystallization buffer (20 mM HEPES pH 7.5, 100 mM NaCl, 1 mM DTT) and mixed 1:1 with well solution containing 1.4–1.6 M sodium malonate pH 6.5–7.0. Large crystals grown over the course of two weeks were cryoprotected in 2.5 M (total) sodium malonate and flash-frozen in liquid nitrogen. Diffraction data were collected at the Advanced Light source beamline 8.3.1 (support statement below). Data were indexed and reduced with XDS [88], scaled with AIMLESS [89], and converted to structure factors with TRUNCATE [90] (S1 Table). The structure was determined by molecular replacement in PHASER using the structure of HIM-3 (PDB ID 4TRK) [19] as a search model. An initial model was rebuilt in COOT [91] and refined in phenix.refine [92] using positional and individual B-factor (anisotropic for protein atoms, isotropic for water molecules) refinement with riding hydrogens. Structure figures were generated in PyMOL version 2.0 (Schrödinger, LLC).

## Supporting information

**S1 Fig. *htp-3^H96Y* does not affect meiotic progression. A.** Full length representative germlines stained with DAPI (magenta) and DSB-1 (green). Yellow dashed line represents region of DSB-1 positive nuclei. Scalebar indicates 20 microns. **B.** Quantification of DSB-1 positive meiotic nuclei for wildtype (n = 1961) and *htp-3 ^H96Y* (n = 1938) mutant strains. NS indicates non-significant. **C.** Quantification of DSB-1 positive meiotic nuclei for *syp-1* (n = 1951) and *htp-3 ^H96Y*;*syp-1* (n = 1847) mutants. NS indicates non-significant. Statistical significance was assessed using two-tailed Fisher's exact tests. Error bars represent 95% confidence intervals.
(PDF)

**S2 Fig. HIM-3^R93Y localizes to meiotic chromosomes. Top:** Mid-pachytene nuclei from wildtype and *him-3^R93Y* germlines stained with DAPI (magenta) and antibody against HIM-3 (green). **Bottom:** Mid-pachytene nuclei from wildtype and *him-3^R93Y* germlines stained with HIM-3 (white). Scalebar indicates 5 microns.
(PDF)

**S3 Fig. *him-3^R93Y* does not affect meiotic progression. A.** Full length representative germlines stained with DAPI (magenta) and antibody against DSB-1 (green). Yellow dashed line represents region of DSB-1 positive nuclei. Scalebar indicates 20 microns. **B.** Quantification of DSB-1 positive meiotic nuclei for wildtype (n = 1981) and *him-3^R93Y* (n = 1918) mutant strains. NS indicates non-significant. **C.** Quantification of DSB-1 positive meiotic nuclei for *syp-1* (n = 1918) and *him-3^R93Y*;*syp-1* (n = 1744) mutants. NS indicates non-significant. Statistical significance was assessed using two-tailed Fisher's exact tests.
(PDF)

**S4 Fig. HTP-1^G97T localizes to meiotic chromosomes. Top:** Mid-pachytene nuclei from wild-type, *htp-1^G97T* and *htp-1^G97T htp-2* germlines stained with DAPI (magenta) and antibody against HTP-1/2 (green). **Bottom:** Mid-pachytene nuclei from wildtype, *htp-1^G97T* and *htp-1^G97T htp-2* germlines stained with HTP-1/2 (white). Scalebar indicates 5 microns.
(PDF)

**S5 Fig. HIM3^R93Y does not affect HIM-3's protein stability. A.** Coomassie stained SDS-PAGE gels of purified HIM-3 and HIM-3^R93Y. **B.** Stability curve for purified wildtype HIM-3 (blue) and HIM-3^R93Y (green).
(PDF)

**S6 Fig. *him-3^R93Y* and *htp-1^G97T* mutants have functional meiotic checkpoints.** Quantification of the average number of apoptotic nuclei per germline. NS indicates non-significant and **** indicates p-value <0.0001. Statistical significance was assessed using two-tailed Student t-tests. Error bars indicate 2X standard error of the mean (SEM).
(PDF)

**S1 Table. Data collection and refinement statistics.** Table with data collection and refinement statistics for Fig 7. Explanations of equations listed below the table.
(PDF)

**S2 Table. Number of nuclei assayed for each genotype in each zone for Figs 3, 4 and 5.**
(PDF)

**S1 Data. Raw data underlying the graphs in the figures.**
(XLSX)

## Acknowledgments

We would like to thank Abby Dernburg, Monique Zetka and Anne Villeneuve for valuable strains and reagents. We would also like to thank the members of the Bhalla lab as well as Brandt Warecki and Alice Devigne for careful review of the manuscript.

## Author Contributions

**Conceptualization:** Anna E. Russo, Kevin D. Corbett, Needhi Bhalla.

**Formal analysis:** Anna E. Russo, Stefani Giacopazzi, Alison Deshong, Kevin D. Corbett, Needhi Bhalla.

**Funding acquisition:** Kevin D. Corbett, Needhi Bhalla.

**Investigation:** Anna E. Russo, Stefani Giacopazzi, Alison Deshong, Malaika Menon, Valery Ortiz, Kaori M. Ego.

**Methodology:** Anna E. Russo, Stefani Giacopazzi, Alison Deshong, Kevin D. Corbett, Needhi Bhalla.

**Project administration:** Kevin D. Corbett, Needhi Bhalla.

**Supervision:** Kevin D. Corbett, Needhi Bhalla.

**Writing – original draft:** Anna E. Russo, Kevin D. Corbett, Needhi Bhalla.

**Writing – review & editing:** Anna E. Russo, Stefani Giacopazzi, Alison Deshong, Malaika Menon, Valery Ortiz, Kevin D. Corbett, Needhi Bhalla.

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
