## [Decision Letter · Decision Letter 0]

24 Jan 2023

Dear Needhi,

Thank you very much for submitting your Research Article entitled 'The conserved AAA ATP-ase PCH-2 distributes its regulation of meiotic prophase events by remodeling multiple meiotic HORMADs in C. elegans' to PLOS Genetics.

The manuscript was fully evaluated at the editorial level and by independent peer reviewers. The reviewers appreciated the attention to an important topic but identified some concerns that we ask you address in a revised manuscript. For example, the reviewers have some concerns about over-interpretation of the data, particularly since no biochemical data are provided to show an interaction between PCH-2 and the HORMADS. Additionally, reviewers have requested more details regarding sample sizes, replicates, and other aspects of the quantitative analysis. We therefore ask you to modify the manuscript according to the review recommendations. Your revisions should address the specific points made by each reviewer, and most can be addressed through changes to the text of the manuscript. 

Yours sincerely,

Paula E. Cohen

Academic Editor

PLOS Genetics

Gregory P. Copenhaver

Editor-in-Chief

PLOS Genetics

Reviewer's Responses to Questions

**Comments to the Authors:**

Reviewer #1: Through elegant genetic experiments, this work describes the role of PCH-2 in regulating the three essential HORMAD proteins at different stages of meiosis in C. elegans. Using a combination of pch-2 null mutants and htp-3/him-3/htp-1 loss of function mutants, the authors convincingly show an important role for PCH-2 regulation of HTP-3 during homolog pairing and synapsis, HIM-3 in recombination, and HTP-1 in meiotic progression. The authors should be commended on an extremely well written and clear manuscript- the rationale for each experiment was well described and the conclusions clearly stated. This is not trivial when working with complex genetic experiments and structural biology. In addition, the combination of genetics, cell biology, and structural biology/biochemistry is a great strength of this manuscript. My major suggestions for improving the manuscript primarily focus on strengthening the biochemical connection between PCH-2 and the HORMADs, and validating the stability of the HORMAD mutant proteins in vivo.

Major Points:

1. The experiments using a combination of mutants to show genetic interactions between PCH-2 and HTP-3/HIM-3/HTP-1 are very well done and convincingly show that PCH-2 must be involved in a pathway regulating these HORMADs. However, the major conclusion and model from this work proposes that PCH-2 directly binds to these HORMADs and changes their conformation. The Introduction and Discussion nicely justify this model: PCH-2 orthologs have been shown to perform this function in other systems. But the authors also describe how PCH-2 regulation of HORMADs may be different in C. elegans (lines 130-132, 654-657) and how the biochemistry of meiotic PCH-2/HORMADs is less convincing (lines 120-122). The assumption made by the authors that PCH-2 is directly interacting with HORMADs to change their conformation is reasonable based on evidence from other organisms, however, this manuscript would be greatly improved if this could be confirmed in C. elegans. I do not think this is a requirement for publication in PLOS Genetics- like I have stated, this assumption is reasonable based on other work and the author’s results- but I do think the authors should make a reasonable attempt to show an interaction more directly and if they cannot, to report this in the manuscript. Some experiments that may accomplish this include (but are not limited to): Immunoprecipitation of either PCH-2 or the individual HORMADs to show PCH-2/HORMAD is at least in complex together in worm lysate, direct binding experiments between purified PCH-2 and HORMADs, or mutants disrupting the region on PCH-2 that binds to HORMADs (if this region has been identified biochemically in other systems and is conserved). If these experiments are not possible because of technical limitations or could not be performed in a reasonable amount of time, the authors should state this in the manuscript and attempt to temper some of the conclusions accordingly; for example, try to avoid using terms like “acts on” (line 237) or “remodels” (line 139) when reporting conclusions if an interaction is only assumed and not directly shown.

2. The conclusion from Figure 1 that the localization of PCH-2 relative to the central element is consistent with PCH-2 interacting with HORMADs would be strengthened if direct co-localization between PCH-2 and HTP-3/HIM-3/HTP-1 could be shown. In addition, this would help support the conclusion of an interaction between PCH-2 and the HORMADs, as I discussed in Major Point 1. I fully appreciate that this is not always possible based on the availability of antibodies (ie if the two antibodies were raised in the same animal), but from reading the methods, it appears that a combination of PCH-2 (rabbit) and at least HTP-3 (guinea pig or chicken) may be possible. If a combination of PCH-2 and HIM-3/HTP-1 is not possible, this should be stated. In addition, showing co-localization between PCH-2 and the different HORMADs throughout meiotic progression may help inform the model that the HORMADs are acted upon by PCH-2 at different stages of meiosis.

3. The authors either cite previous work or show localization of mutant HTP-3/HIM-3/HTP-1 (Figs S2, S4) to meiotic chromatin as evidence that these mutant proteins are abundant and stable. They also nicely show that mutant HIM-3 is stable in vitro (Figure S5). However, it is still important to show the abundance of these mutant proteins in worm lysate to rule out any dosage effects on the observed phenotypes. It is possible that these mutations make a particular conformation or population (i.e. the diffusive nuclei population mentioned on lines 173-174) unstable, which would lead to a different conclusion than if the protein functions normally except the mutated region. This is especially important for HTP-1 which the authors explain is susceptible to instability. This could be accomplished with a quantitative western blot comparing the relative abundance of these proteins in wild type control worm lysate and mutant worm lysate.

Minor points:

1. Please expand the Methods sub-section “Protein Expression and Purification” to include relevant buffers (lysis buffer, elution buffers, final buffer pure protein is in). This will make it possible to replicate these results. Also, if possible, please include a final gel of the purified protein that was used in the biochemistry experiments (Figs 7 and S5).

2. In lines 270-271, a very brief explanation of why C. elegans have relatively low RAD51 foci numbers compared to mammals would be appreciated for readers who are less familiar with the worm system. In mammalian spermatocytes, there are hundreds of RAD51 foci per nucleus, so it is surprising that worm oocytes only contain ~2 per nucleus.

3. I am confused by the error bars in Figure 4K- the text states that the authors observe no univalents in wild type and pch-2; him-3R93Y, but there are significant error bars. Perhaps my understanding of statistics is failing me, but consider including an explanation?

4. Is it possible to show a full structure of HTP-1 to include with Figure 2E, like what is shown in Figs 2C and D? It would be nice to have the inset in context with the whole protein.

5. The peptide binding assays shown in Fig 7 are beautifully done. Is it possible to expand this assay with additional purified HORMADs or with more closure motif peptides? I appreciate that this mechanism is likely generalizable but more examples would strengthen this conclusion.

6. On lines 662-663 of the Discussion, the authors mention PCH-2 may reduce overall levels of HORMADs to modulate meiotic events. Could a Western blot from the pch-2 mutants show this change in HORMAD abundance?

7. Please expand on the “IF, Antibodies and Microscopy” Methods sub-section. More details including incubation times, relevant buffers, fixation times, etc. This would help others replicate these results.

Reviewer #2: In this paper, Russo and colleagues show that the AAA+ ATPase PCH-2 exerts its regulatory control over several critical events in meiotic prophase in C. elegans through its interaction with different HORMA-domain proteins of the chromosome axis. Mutations in analogous positions in the HORMA domains of three proteins (HTP-3, HIM-3, and HTP-1) are found to suppress distinct phenotypes of pch-2 single mutants, leading to the conclusion that genetic interactions between PCH-2 and these HORMA proteins are important in controlling distinct processes (pairing, synapsis, crossover recombination, and meiotic progression) in meiotic prophase. Since these mutations lower the affinity of HORMA domain proteins for their closure motifs, while the assumed activity of PCH-2 is to convert the closed form of HORMA proteins to the open form, these experiments strongly suggest the idea that PCH-2 exerts control on the above events by remodeling HORMA domain proteins. These results are contrasted with the known roles of PCH-2 orthologs in other species and their biological significance is discussed in light of several particular aspects of C. elegans meiosis (such as pairing being independent of DNA breaks). This paper sheds new light on possible functions of PCH-2 and opens a new avenue toward further mechanistic understanding of the critical roles played by HORMA-domain proteins of the meiotic chromosome axis.

Major comments:

1. The genetic evidence provided for HORMAD<->PCH-2 interaction is strong for the most part, but to interpret some of the figures it will be necessary to add a full description of the quantitation method and statistical analysis. For example, in Figure 6B, the mean is shown, but there is no indication of how many gonads were examined, which cells from the gonad were considered, and how the percentages of DSB-1-positive cells from each gonad were aggregated into the Fisher's exact test. Especially in measurements with small effect size (for example, 6B showing meiotic progression in htp-1(G97T) vs pch-2;htp-1(G97T), 81% vs 77% DSB-1 positive), a full accounting of the quantitative analysis will lend more support to the conclusions.

2. The number of biological/technical replicates should be listed for each figure, and the data underlying the summary statistics made available as per PLOS policy.

3. There are some conclusions made in the text that appear to be based either on observations without statistical significance, or that have not been assessed for significance, in the corresponding Figure (for example, from line 408: "we found that there was an increase in the percent of nuclei with univalents in pch-2 single mutants compared to wildtype controls…" but the figure 4K says it's not significant). This is an issue for several of the graph panels of Figures 3, 4, and 5. If there is little statistical significance to some results, then the conclusions should be given appropriate caveats.

4. I also have a question about a point of logic in the paper, concerning how the apparently disparate genetic interactions of pch-2 with htp-3(H96Y) and him-3(R93Y) are treated. The fact that htp-3(H96Y) rescues the early synapsis phenotype of pch-2 mutants has a straightforward interpretation: the H to Y mutation makes HTP-3 more readily transition from a closed to open state (which pch-2 is normally required for). If this were the mode of action of PCH-2 on HIM-3, however, then the him-3R93Y single mutation should not phenocopy pch-2 mutations (Fig. 4I), but be more similar to wildtype, as is the case with the single htp-3(H96Y) mutation. Currently the text says that "these two alleles genetically suppress each other in the context of crosover recombination" (line 414), but in the Discussion, the mode of PCH-2 action on HTP-3 and HIM-3 are not distinguished (line 725-6). If pch-2 works by opening HIM-3, and the mutant has lower affinity for closure motifs (i.e. is more likely to adopt the open confirmation), it is not clear to me how these alleles could suppress each other. Some plausible explanation of this should be offered in the Discussion.

Minor points that should be clarified or considered:

- In lines 97–113, from the description given it is not clear to me how Hop1 in solution is supposed to be conceived -- in the "closed" conformation does it bind to a binding partner (line 97:"In the closed conformation, a peptide on a binding partner called the 'closure motif' binds the core of the HORMAD’s HORMA domain.") or to itself?

- Line 232: htp-3(H96Y) is said here to have "accelerated desynapsis", but desynapsis looks delayed instead of accelerated in Fig. 3E (pos 5,6 have higher % of nuclei with full synapsis than WT). Further, the reference given for previous report of this is #42, but ref. 42 does not refer to this.

- Line 393: the text says "Less than 6" (COSA-1 foci) but Fig 4I says just "5"; please clarify whether the measurements here and in Fig. is for exactly 5 foci or "5 or fewer" foci.

- Please address the seeming discrepancy between the observations that RAD-51 foci are higher in the him-3R93Y mutant (Fig. 4G), while COSA-1 foci are lower (Fig. 4I) — i.e., why do higher DSB levels coexist with loss of CO assurance?

- Finally, while the paper views HORMA domain protein function mainly through the lens of PCH-2 interaction, many of the non-PCH-2-related phenotypes seen in these novel point mutants are quite interesting in their own right and should not be deemphasized, as they provide more evidence for the idea that distinct meiotic functions are delegated to distinct HORMA domain proteins. The authors might wish to consider summarizing the PCH-2-dependent and -independent roles found for each HORMA protein in a table or chart.

Reviewer #3: In the study titled “The conserved AAA-ATPase PCH-2 distributes its regulation of meiotic prophase events by remodeling multiple meiotic HORMADs in C. elegans” Russo et al. show that pch-2, the pachytene checkpoint homolog, genetically interacts with specific alleles of the three HORMADs (point mutations in the HORMA domain), htp-3, him-3 and htp-1 in different contexts during meiotic prophase in C. elegans. The authors determine that the specific alleles tested for each of the HORMAD proteins result in unique genetic interactions with pch-2. Biochemically, the authors test whether these genetic interactions affect the binding ability of the HORMAD proteins by assaying HIM-3R93Y's binding to the closure motif on HTP-3 in vitro. The authors conclude that PCH-2 remodels each of the individual HORMADs to regulate distinct meiotic functions and propose that PCH-2 remodels meiotic HORMADs from a closed to unbuckled conformation to control the progress of pairing, synapsis, meiotic recombination, and cell cycle progression.

While the ability to determine the specific role of PCH-2 in mediating its vast set of functions during meiotic prophase is important, the data presented in the manuscript falls short of the conclusion that PCH-2 remodels or regulates the HORMAD proteins. Instead, what the authors can conclude is that there is a genetic interaction between the HORMAD protein alleles and pch-2, which may very well be indirect in facilitating the final outcomes assayed, since the biochemical assay used in the manuscript does not test the role of PCH-2 in mediating the interaction between the closure motif on HTP-3 and the HIM-3R93Y allele for example. The authors make several over interpretations with very strong conclusions that are not supported by the evidence presented. The conclusions and interpretations should be dialed back to align with the data. Additionally, there is overall a lack of scientific precision in methods presented, legends, and some experiments lack of control groups (e.g., htp-1 G97T on meiotic progression, no pch-2 alone). We provide detailed comments below.

Major comments:

1. Usage of the word remodeling or regulation of HORMADs by PCH-2 is an overstatement and misleading, including in the title of the manuscript. The experiments conducted in the manuscript do not test whether PCH-2 remodels the three HORMAD domain mutant alleles. Instead, they test whether pch-2 mutant genetically interacts with these alleles. As stated above, the genetic interactions may well be indirect, which the authors acknowledge in the Discussion portion of the manuscript: Line 662: “PCH-2 either reduces overall levels of HORMADs or increases their association-dissociation dynamics to modulate these key events”; Line 673-675: “An attractive but still speculative extension to this model is that PCH-2’s remodeling of meiotic HORMADs destabilizes HORMAD-partner protein interactions”; Line 718-719, “PCH-2’s remodeling of HTP-3 and HIM-3 may indirectly result in the removal, and remodeling, of HTP-1, contributing to the regulation of meiotic progression”. The conclusions and title should be based on direct rather than speculative analysis.

2. Methods section requires more details. For example, the number of replicates for each experiment needs to be provided. Method used for analysis of number of germlines per experiment states that “a minimum of three germlines” were used per genotype; three is a very small number, leading to the question of how the statistics were generated. Number of germ cells analyzed for each of the assays of pairing, synapsis, DSB, etc need to be presented per experiment. Details on measurement of percent viability and percent male self-progeny analysis needs to be provided. Overall, the details provided in the methods section are minimal which renders the manuscript scientifically less rigorous in its current state.

3. Line 161: The overlapping nature of the SYPs and PCH-2 immunostaining has been published by the authors previously. Since the goal of the study is to test whether PCH-2 can reside on or close to axial elements of synapsed chromosomes and remodel meiotic HORMADs on chromosomes, analysis of localizations between the HORMADs with PCH-2, rather than the SYPs, may be more informative.

4. Counting of RAD-51 foci (Fig. 3G, H and others): Almost all of the nuclei shown in the representative image (genotype and region not labeled) present with more than 3 foci, yet none of the groups on the graphs represent foci >2.5. This data needs to be better reconciled with the images shown.

5. In Line 271, the authors state that there was “an average of 2 per nuclei” yet the figure does not show this. Similarly the error bars (SEM) do not capture the spread of the data points. In addition, previously, the authors showed ~4 foci of RAD-51 at zone 4 for wild type germlines (Fig. 4B, ref 3), while in the analysis in the current study this number seems to be far lower in the graphs. This is an important difference considering the very subtle difference in wild type and mutant germlines, a change in 2 foci/nuclei in wild type could cause error in data interpretation.

6. Percent DSB-1 positive meiotic nuclei: i) There is no pch-2 single group to compare with. And ii) the difference between htp-1G97T and pch-2; htp-1G97T shown in the graph (Fig 6B) is very subtle. Given the small difference in the data, a greater number of germlines will need to assayed to assess robust statistical significance.

Minor comments:

Figure 2 should follow Figure 1 and be proceeded by Figure 3 in the writing. As written, Figure 2 appears after Figure 3.

Fig. 1 B: Please provide the genotype

Fig. 3B, D, G, I; Fig 4A, C, F, H, J; Fig. 5A, C, E, G, I: Genotype and zone numbers need to be labeled. The authors do not present representative images of defects from all the genotypes assayed and instead use one genotype on the left panel for each one of these figures. Unfortunately, this form of representation does not add any value. If the point of the single image is to show how the germ cell defect appears and how the counting is done, showing it once in the beginning would serve the purpose.

Line 187: “Having established that PCH-2’s localization is consistent with the ability to remodel meiotic HORMADs on meiotic chromosomes, we looked for genetic interactions that might support this possibility.”- is not accurate, since the data only suggests a possible proximity between PCH-2 and HORMADs, which at the resolution tested is not indicative of remodeling.

Line 237: “interacts with” rather than “acts on” would be more accurate.

Line 634: needs a reference.

Line 638: needs a reference.

Line 696-698: This statement is inaccurate. The role of HTP-1 in regulating meiotic progression is well documented in the ref [53 and 72]. In addition, in the ref 72, the authors demonstrate that phosphorylation of HTP-1 regulates synapsis formation and maintenance which in turn regulates meiotic progression. Thus, HTP-1-mediated effect on synapsis is not “indirect” and is the consequence of defects in synapsis formation and maintenance due to lack of HTP-1 phosphorylation.

Line 771-772: Please see the above comment.

Line 719: Omit the comma.

Line 1429: Since the manuscript does not address how PCH-2 may act to remodel the different meiotic HORMADs the title of the model figure should be altered to more accurately reflect data presented on the interactions of pch-2 with the hormads.

**Have all data underlying the figures and results presented in the manuscript been provided?**

Reviewer #1: **No: **Numerical data underlying graphs not provided.

Reviewer #2: **No: **Numerical data for summary statistics has not been provided (plots/graphs in Figures 3,4,5,6,S1,S3,S6)

Reviewer #3: Yes

PLOS authors have the option to publish the peer review history of their article (what does this mean?). If published, this will include your full peer review and any attached files.

Reviewer #1: No

Reviewer #2: No

Reviewer #3: No

---

## [Editor Report · Decision Letter 1]

21 Mar 2023

Dear Needhi,

We are pleased to inform you that your manuscript entitled "The conserved AAA ATPase PCH-2 distributes its regulation of meiotic prophase events through multiple meiotic HORMADs in C. elegans" has been editorially accepted for publication in PLOS Genetics. Congratulations!

Yours sincerely,

Paula E. Cohen

Academic Editor

PLOS Genetics

Gregory P. Copenhaver

Editor-in-Chief

PLOS Genetics

Comments from the reviewers (if applicable):

**Data Deposition**

http://datadryad.org/submit?journalID=pgenetics&manu=PGENETICS-D-22-01314R1

**Press Queries**

---

## [Editor Report · Acceptance letter]

11 Apr 2023

PGENETICS-D-22-01314R1 

The conserved AAA ATPase PCH-2 distributes its regulation of meiotic prophase events through multiple meiotic HORMADs in C. elegans 

Dear Dr Bhalla, 

We are pleased to inform you that your manuscript entitled "The conserved AAA ATPase PCH-2 distributes its regulation of meiotic prophase events through multiple meiotic HORMADs in C. elegans" has been formally accepted for publication in PLOS Genetics! Your manuscript is now with our production department and you will be notified of the publication date in due course.

With kind regards,

Anita Estes

PLOS Genetics

On behalf of:
